# Multi-model Ensemble Conformal Prediction in Dynamic Environments

**Erfan Hajihashemi**
Department of Electrical Engineering & Computer Science
University of California, Irvine
`ehajihas@uci.edu`

**Yanning Shen**[*]
Department of Electrical Engineering & Computer Science
University of California, Irvine
`yannings@uci.edu`

## Abstract

Conformal prediction is an uncertainty quantification method that constructs a prediction set for a previously unseen datum, ensuring the true label is included with a predetermined coverage probability. Adaptive conformal prediction has been developed to address data distribution shifts in dynamic environments. However, the efficiency of prediction sets varies depending on the learning model used. Employing a single fixed model may not consistently offer the best performance in dynamic environments with unknown data distribution shifts. To address this issue, we introduce a novel adaptive conformal prediction framework, where the model used for creating prediction sets is selected 'on the fly' from multiple candidate models. The proposed algorithm is proven to achieve strongly adaptive regret over all intervals while maintaining valid coverage. Experiments on real and synthetic datasets corroborate that the proposed approach consistently yields more efficient prediction sets while maintaining valid coverage, outperforming alternative methods.

## 1   Introduction

Most machine learning algorithm designs aim to enhance label prediction accuracy. Nevertheless, a significant challenge persists as many models demonstrate limitations in predicting labels with high certainty, falling short of achieving the desired levels of accuracy and other critical evaluation metrics. In applications such as medical diagnosis, it is sometimes more efficient to predict a subset of labels rather than a single label [Levy et al., 2021, Straitouri et al., 2023]. This necessitates predicting a set of candidate labels with a valid coverage probability, rather than limiting to a single label.

One of the most widely used frameworks for set prediction is conformal prediction [Vovk et al., 2005]. Conventional conformal prediction algorithms can achieve the desired coverage assuming the exchangeability of data [Balasubramanian et al., 2014]. However, in many real-world online problems, the distribution of data shifts over time, making the exchangeability assumption no longer applicable. Consequently, adaptive conformal prediction algorithms have been developed [Gibbs and Candes, 2021], where prediction sets are constructed in a time-varying manner. Despite these advancements, the efficiency (e.g., prediction set size or regret) of previous conformal prediction methods in online settings with distribution shifts heavily depends on the model employed, and

---

[*]Corresponding author

38th Conference on Neural Information Processing Systems (NeurIPS 2024).

a single model may not consistently perform well across various distribution shifts. Creating an efficient prediction set size is as important as obtaining the desired coverage. Trivial cases may arise where the prediction set alternates between an empty set and the full set of labels, with the full set being created with a probability equal to the desired coverage probability, and empty sets otherwise. This approach achieves the desired coverage but leads to impractical prediction sets [Bhatnagar et al., 2023]. To address this limitation, our proposed algorithm incorporates multiple learning models simultaneously. It dynamically selects the suitable model based on the performance of each model with the most recently received data.

**Related work:** Conformal prediction [Vovk et al., 2005, Shafer and Vovk, 2008, Vovk, 2015] is an effective method for uncertainty quantification that has been widely used to predict a set of candidate labels for income data. It treats the learning model as a black box and provides a prediction set for new test data. Conformal prediction is applicable in both Classification [Ding et al., 2024, Shi et al., 2013, Romano et al., 2020] and Regression [Romano et al., 2019, Papadopoulos et al., 2011, Boström et al., 2017] problems. In dynamic environments where data distribution shifts over time, using vanilla conformal prediction algorithms may not lead to the desired coverage performance. To cope with this challenge, conformal prediction in dynamic environments has been studied recently. [Tibshirani et al., 2019] explored conformal prediction for dynamic settings using a reweighting approach, but their method requires prior information about the data dependency structure; [Barber et al., 2023] resolved this dependency by requiring weights to be fixed. Incorporating time-varying coverage probability was introduced in [Gibbs and Candes, 2021], but determining the appropriate step size remains a significant challenge. One way to address dynamic environments is adopting learning with expert advice [Cesa-Bianchi et al., 1997, Vovk, 1995, Littlestone and Warmuth, 1994]. [Zaffran et al., 2022] suggested that tuning the step size based on expert (base learner) aggregation could be an effective strategy; however, this method causes each expert to receive equal impact from all historical data, which makes the algorithm unable to adapt to sharp distribution shifts. Furthermore, [Gibbs and Candès, 2022] introduced an approach that employs multiple experts, each assigned a distinct step size from a pool of candidate sizes, resulting in varying coverage probabilities at each time $t$. While [Gibbs and Candès, 2022] successfully demonstrated adaptive regret across time intervals of a fixed width, [Bhatnagar et al., 2023] points out that this approach fails to achieve suitable regret across varying widths of arbitrary time intervals simultaneously. [Bhatnagar et al., 2023] addressed this limitation by establishing strongly adaptive regret [Daniely et al., 2015] across any arbitrary time interval width. Their proposed methodology assigns a specific time interval to each expert. Despite achieving sublinear strongly adaptive regret, the efficacy of this approach depends on the hyper-parameter selection that determines each expert's lifetime. The proposed method in our study also employs experts who operate within specific time intervals. However, each expert consists of multiple learning models, aiming to select the appropriate model according to the specific data distribution during its operation, resulting in more efficient prediction sets.

**Contributions.** Overall, our contributions can be summarized as follows:
**I)** We introduce a novel adaptive conformal prediction algorithm, **S**trongly **A**daptive **M**ultimodel Ensemble **O**nline **C**onformal **P**rediction (SAMOCP), designed for dynamic environments with unknown distribution shifts. This algorithm incorporates multiple models and dynamically selects a model based on its performance in previous time steps.
**II)** We demonstrate that SAMOCP exhibits strongly adaptive regret for any arbitrary time interval while ensuring valid coverage.
**III)** Through experimental tests on classification tasks subject to distribution shifts, we demonstrate that SAMOCP outperforms existing methods by constructing more efficient prediction sets while also achieving a coverage probability closely aligned with the target value.

## 2   Preliminaries

This section explains standard conformal prediction and adaptive online conformal predictions, where data is collected sequentially. We begin with outlining standard conformal prediction. Given a miss coverage probability $\alpha$, a learning model $m$ , a historical dataset $\{(X_\tau, Y_\tau^{\text{true}})\}_{\tau=1}^{t-1}$, and a new data $X_t \in \mathcal{X}$, the objective is to construct a prediction set $C_\alpha^m(X_t) \subseteq \mathcal{Y} := \{1, 2, \ldots, K\}$, where $K$ denotes the total number of classes, such that $C_\alpha^m(X_t)$ contains the true label $Y_t^{\text{true}}$ with probability $1 - \alpha$. In the online setting, historical dataset is updated to $\{(X_\tau, Y_\tau^{\text{true}})\}_{\tau=1}^{t}$ at the end of each time $t$ when true label $Y_t^{true}$ for input data $X_t$ is observed. In this scenario, conformal prediction treats

the historical dataset as a calibration dataset, which is utilized to determine whether a candidate label $Y \in \mathcal{Y}$ should be included in the prediction set. Consequently, in an online manner, conformal prediction relies on an evolving calibration dataset, which contains all the historical data 'on the fly' to decide which candidate labels should be included in the prediction set. Non-conformity scores $\{S^m(X_\tau, Y_\tau^{\text{true}})\}_{\tau=1}^{t-1}$ are introduced. Specifically, each non-conformity score $S^m(X_\tau, Y_\tau^{\text{true}})$ assesses the disagreement between the ground-truth label $Y_\tau^{\text{true}}$ and predicted label $\hat{f}^m(X_\tau)$. Upon obtaining a new datum $X_t$, the standard conformal prediction algorithm constructs the prediction set for $X_t$ as $C_\alpha^m(X_t) = \{Y \in \mathcal{Y} \mid S^m(X_t, Y) \leq \hat{q}_\alpha^m\}$, where the threshold $\hat{q}_\alpha^m$ is obtained as

$$\hat{q}_\alpha^m = Quantile\left(\frac{\lceil t(1-\alpha) \rceil}{t-1}, \{S^m(X_\tau, Y_\tau^{true})\}_{\tau=1}^{t-1}\right). \tag{1}$$

The Quantile function sorts all non-conformity scores of the historical data and then identifies the $\lceil t(1-\alpha) \rceil$th smallest score as $\hat{q}_\alpha^m$. Note that $1-\alpha$ is fixed, hence conformal prediction cannot readily cope with potential data distribution shifts in dynamic environments. Adaptive conformal prediction algorithms have been developed to address this issue, allowing the miss coverage probability to vary at each time $t$ and thereby enabling the algorithm to dynamically adapt to potential shifts in the distribution. In such a scenario, $\hat{q}_\alpha^m$ can be obtained by replacing $\alpha$ with $\alpha_t$ in (1). Then at each time step $t$, $\alpha_t$ is updated after observing $Y_t^{true}$.

Recent studies on online conformal prediction with distribution shifts have incorporated adaptive miss coverage probabilities to address dynamic environments. However, methods based on a single learning model may not achieve consistently reliable performance in dynamic environments. This underscores the necessity for employing multiple models and adaptive strategies to determine the appropriate model for each time $t$. To this end, in this work, we introduce a novel adaptive multi-model online conformal prediction algorithm designed to identify the suitable learning model at each time $t$ within dynamic environments. At time slot $t$, the goal is to construct a prediction set for the new data $X_t$, based on the historical dataset $\{(X_\tau, Y_\tau^{\text{true}})\}_{\tau=1}^{t-1}$, such that the true label is included in the prediction set with probability $1-\alpha$. The proposed algorithm for dynamic settings achieves strongly adaptive regret while ensuring valid coverage.

## 3 Methodology

In this section, two adaptive algorithms are developed for static and dynamic environments respectively. Subsection 3.1 develops the **M**ultimodel Ensemble **O**nline **C**onformal **P**rediction (MOCP) algorithm to identify the suitable learning model among $M$ distinct candidates in a static environment. Subsequently, in Subsection 3.2, we propose SAMOCP, an adaptation of MOCP tailored for dynamic environments with unknown distribution shifts.

### 3.1 Multi-model Conformal Prediction in Static Environments

Note that the non-conformity score $S^m(X_\tau, Y_\tau^{true})$ depends on the learning model. Such dependency leads to a model-specific ordering of non-conformity scores, yielding different prediction sets for each model. After observing $Y_t^{true}$, the adaptive miss coverage probability $\alpha_t$ must be updated for time $t+1$ to cope with distribution shifts effectively. Given that different models achieve different prediction sets, assigning and updating the same $\alpha_t$ for different models would be inadequate. Instead, at each time $t$, we assign a specific miss coverage probability to each model $m \in [M]$, denoted as $\alpha_t^m$, and update it based on the corresponding prediction set. Consequently, for $M$ learning models, there are $M$ candidates for miss coverage probability $\alpha_t$ at each time $t$. Each candidate is updated according to a distinct rule. These $M$ update rules operate in parallel, with each one updating the corresponding miss coverage probability upon observing the true label. Next, the update procedure for $\alpha_t^m$ will be examined, followed by a detailed explanation of how each instance of MOCP selects the appropriate miss coverage probability from $M$ distinct options at each time step.
To update miss coverage probability $\alpha_t^m$, we adopt the pinball loss [Koenker and Bassett, 1978], which can be written as

$$L(\bar{\alpha}_t^m, \alpha_t^m) = \alpha(\bar{\alpha}_t^m - \alpha_t^m) - \min\{0, \bar{\alpha}_t^m - \alpha_t^m\}, \tag{2}$$

where

$$\bar{\alpha}_t^m = \sup\{\tilde{\alpha} : Y_t^{true} \in C_{\tilde{\alpha}}^m(X_t)\} \tag{3}$$

is the best possible value of miss coverage probability for model $m$ at time $t$ which constructs the smallest prediction set that covers $Y_t^{true}$. The miss coverage probability $\alpha_{t+1}^m$ can be updated via SF-OGD [Orabona and Pál, 2018] as

$$\alpha_{t+1}^m = \alpha_t^m - \eta \frac{\nabla_{\alpha_t^m} L(\bar{\alpha}_t^m, \alpha_t^m)}{\sqrt{\sum_{\tau=1}^t \|\nabla_{\alpha_\tau^m} L(\bar{\alpha}_\tau^m, \alpha_\tau^m)\|_2^2}}, \tag{4}$$

where $\eta$ is the learning rate and

$$\nabla_{\alpha_t^m} L(\bar{\alpha}_t^m, \alpha_t^m) = \mathbb{I}[\bar{\alpha}_t^m < \alpha_t^m] - \alpha = err_t^m - \alpha, \tag{5}$$

with $err_t^m := \mathbb{I}[Y_t^{true} \notin C_{\alpha_t^m}^m]$ equals 1 if the predicted set does not contain the true label $Y_t^{true}$, and 0 otherwise. According to the updating rule outlined in equation (4), the adjustment of $\alpha_t^m$ at each time $t$ is governed by the $\nabla_{\alpha_t^m} L(\bar{\alpha}_t^m, \alpha_t^m)$, as detailed in equation (5). When $err_t^m = 1$, it signals that the coverage probability $1 - \alpha_t^m$ is too small, resulting in a prediction set that can not encompass $Y_t^{true}$. Consequently, there's a necessity to enlarge the coverage probability, effectively achieved by reducing $\alpha_t^m$, which would be facilitated by (4); given that the denominator in the second term is always positive and the gradient will be positive in this scenario. On the other hand, when $\bar{\alpha}_t^m > \alpha_t^m$, $1 - \alpha_t^m$ leads to a prediction set that covers $Y_t^{true}$ but also includes unnecessary labels $\mathcal{Y}' := \{Y' \in \mathcal{Y} \mid \hat{q}_{\bar{\alpha}_t^m}^m < S^m(X_t, Y') \leq \hat{q}_{\alpha_t^m}^m\}$. In such cases, optimization necessitates increasing $\alpha_t^m$ to avoid including unnecessary labels and output a more efficient prediction set. This adjustment is facilitated by the update rule (4).

Additionally, the weight $w_t^m$ is assigned to each model $m \in [M]$, which influences the selection of its corresponding miss coverage probability $\alpha_t^m$. MOCP learns which model to select over time based on the performance of each model over previous time steps, as reflected in $w_t^m$. The algorithm updates the weight associated with each model after revealing the true label $Y_t^{true}$. This update is performed with respect to the loss function of the corresponding miss coverage probability. Specifically, $w_t^m$ can be updated by

$$w_{t+1}^m = w_t^m \exp\left(-\epsilon L\left(\bar{\alpha}_t^m, \alpha_t^m\right)\right), \tag{6}$$

where $0 < \epsilon < 1$ is the step size. At each time $t$, upon receiving new data $X_t$, MOCP first calculates the normalized weights, denoted as $\{\bar{w}_t^m\}_{m=1}^M$. For any $m \in [M]$, $\bar{w}_t^m = \frac{w_t^m}{\sum_{j=1}^M w_t^j}$ ensures that $\bar{w}_t^m \in [0, 1]$ and represents the likelihood of selecting miss coverage probability $\alpha_t^m$. Then, the algorithm selects miss coverage probability $\alpha_t^{\hat{m}}$, where $\hat{m} \in [M]$, according to PMF $\bar{\boldsymbol{w}}_t := (\bar{w}_t^m)_{m=1}^M$, i.e., each miss coverage probability $\alpha_t^m$ is selected with probability proportional to the corresponding normalized weight $\bar{w}_t^m$. The prediction set for $X_t$ is constructed according to the threshold in (1), by replacing $\alpha$ and $m$ with $\alpha_t^{\hat{m}}$ and $\hat{m}$ respectively. After receiving $Y_t^{true}$, each weight $w_t^m$ and miss coverage probability $\alpha_t^m$ are updated according to (6) and (4), respectively. This entire process is detailed in Algorithm 1. The MOCP algorithm achieves a runtime of $\mathcal{O}(T)$ when the number of models $M$ is constant.

Given that the environment is static, there exists a miss coverage probability that can minimize the loss function for each model $m \in [M]$ over $[T]$, denoted as $\alpha^m$. The best miss coverage probability among $\{\alpha^m\}_{m=1}^M$ can be obtained by

$$\alpha^{m^*} = \underset{\{\alpha^m, m \in [M]\}}{\arg\min} \sum_{t=1}^T L(\bar{\alpha}_t^m, \alpha^m) \quad \text{with} \quad \alpha^m = \underset{\alpha_t^m}{\arg\min} \sum_{t=1}^T L(\bar{\alpha}_t^m, \alpha_t^m). \tag{7}$$

The following theorem demonstrates that MOCP achieves sublinear regret (See proof in A.2).

**Theorem 1** *Algorithm 1 achieves the following regret bound in a static environment*

$$\sum_{t=1}^T \sum_{m=1}^M \bar{w}_t^m L(\bar{\alpha}_t^m, \alpha_t^m) - \sum_{t=1}^T L(\bar{\alpha}_t^{m^*}, \alpha^{m^*}) \leq \sqrt{T} \left(\frac{(1+2\eta)^2}{2\eta} + \frac{\eta}{2\alpha} + \ln M + (1+\eta)^2\right). \tag{8}$$

### 3.2 Multi-model Ensemble Conformal Prediction In Dynamic Environments

Through Algorithm 1, we demonstrated how to select the suitable miss coverage probability of a model that has achieved lower loss compared to other models over previous time steps. However,

**Algorithm 1** Multi-model Ensemble Online Conformal Prediction (MOCP)

**Require:** $\alpha \in [0, 1]$, learning rate $\eta \geq 0$, and step size $\epsilon \in (0, 1)$.

$\quad w_1^m \leftarrow \frac{1}{M}$ {Initialization of weight for each model}

$\quad \alpha_1^m \leftarrow \alpha$ {Initialization of miss coverage probability for each model}

$\quad$**for** $t \in [T]$ **do**

$\quad\quad$ Observe $X_t \in \mathcal{X}$.

$\quad\quad$ Calculate normalized weights by $\bar{w}_t^m = \frac{w_t^m}{\sum_{j=1}^M w_t^j}, \forall m \in [M]$.

$\quad\quad$ Select one of the miss coverage probabilities $\{\alpha_t^m\}_{m=1}^M$ according to PMF $\bar{\boldsymbol{w}}_t = (\bar{w}_t^m)_{m=1}^M$.

$\quad\quad$ Observe true label $Y_t^{true}$ and compute optimal value $\bar{\alpha}_t^m \; \forall m \in [M]$ with (3).

$\quad\quad$**for** $m \in [M]$ **do**

$\quad\quad\quad$ Obtain loss $L(\bar{\alpha}_t^m, \alpha_t^m)$.

$\quad\quad\quad$ Update $w_{t+1}^m$ with (6).

$\quad\quad\quad$ Update $\alpha_{t+1}^m$ with (4).

$\quad\quad$**end for**

$\quad$**end for**

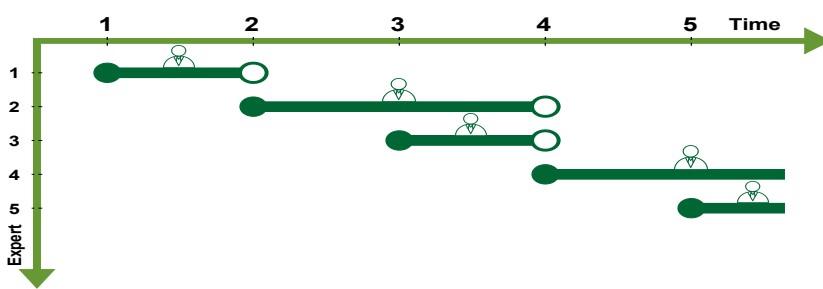

Figure 1: Expert creation over 5 time steps using lifetime formula (9) when $g = 1$. At each time $t$, an expert is created, marked by a filled circle to indicate the start of the activity, and an unfilled circle to denote the end of the expert's activity.

this approach assumes a static environment that does not change over time, which may limit the algorithm's effectiveness in dynamic environments where data distribution shifts occur. In addition, the selection of stepsize $\epsilon$ in (6) critically affects the performance. In environments with unknown distribution shifts, a large $\epsilon$ indicates faster adaptation to abrupt changes, whereas a small $\epsilon$ is more suitable for environments with less variability. Thus, the efficient choice of $\epsilon$ depends on the variability of the environment, which poses a challenge in scenarios with unknown distribution shifts.

To address this limitation, we introduce Strongly Adaptive (SA)MOCP. Specifically, each instance of MOCP is treated as an 'expert', and multiple experts are created at distinct time steps with specific step sizes and lifetimes to cope with potential distribution shifts. At the end of its lifetime, the expert becomes inactive, ensuring that it no longer affects the decision-making process, refer to Figure 1 for an illustration. This strategy prevents outdated experts from contributing to the selection of the suitable miss coverage probability in dynamic environments. Subsequently, SAMOCP dynamically selects the appropriate expert for each time $t$ and utilizes its chosen miss coverage probability to construct the prediction set. The lifetime of each expert is determined by the specific time $t$ at which it was created and hyperparameter $g \in \mathbb{N}$, as [Bhatnagar et al., 2023].

$$\lambda(t) = g \cdot \max_{n \in \mathbb{Z}} \{2^n : t \equiv 0 \mod 2^n\}. \tag{9}$$

The active interval of the expert created at time $t$ is defined as $[t, t + \lambda(t) - 1]$. Consequently, experts considered active at any given time $\tau$ are those whose active intervals include $\tau$. In a dynamic setting with unknown distribution shifts, the best model may vary across different distributions. This variability results in scenarios where, at each time $t$, some active experts might not have adapted to the current data distribution yet, and thus they rely on different models compared to more recently established active experts. In such cases, the miss coverage probability of model $\hat{m} \in [M]$ chosen by expert $n$ at time $t$ is represented as $\alpha_t^{\hat{m}n}$, and its best possible value is denoted as $\bar{\alpha}_t^{\hat{m}n}$. To select the suitable miss coverage probability among different experts, we assigned distinct weights to each,

with specific initialization and step sizes for updates. Specifically, this weight for expert $n$ at time $t$ is denoted as $h_t^n$, and its step size is defined as $\epsilon^n := \min(\epsilon, \frac{\sigma}{\sqrt{\lambda(n)}})$, where $\sigma > 1$ is a constant and $\lambda(n)$ is the lifetime for expert $n$ that obtained by (9). Given that $n$th expert is activated at $t = n$, the initialization and update rule for $h_t^n$ is as follows:

$$h_{t+1}^n = \begin{cases} \epsilon^n & \text{if } t = n - 1 \\ h_t^n \exp\left(-\epsilon^n \cdot r_t^n\right) & \text{if } t \in [n, n + \lambda(n) - 1) \\ 0, & \text{otherwise} \end{cases} \tag{10}$$

where $r_t^n = L(\bar{\alpha}_t^{\hat{m}\hat{n}}, \alpha_t^{\hat{m}\hat{n}}) - L(\bar{\alpha}_t^{\hat{m}n}, \alpha_t^{\hat{m}n})$ represents the loss of the $n$th expert relative to the loss of the learner who selects expert $\hat{n}$. We denote the set of active experts at each time $t$ as $\mathcal{A}(t)$. The learner selects the suitable miss coverage probability among all active experts according to the PMF $\bar{h}_t := (\bar{h}_t^n)_{n \in \mathcal{A}(t)}$, where each $\bar{h}_t^n$ represents the normalized version of the weight $h_t^n$, calculated as $\bar{h}_t^n = \frac{h_t^n}{\sum_{i \in \mathcal{A}(t)} h_t^i}$, ensuring that $\bar{h}_t^n \in [0, 1]$. Algorithm 2 summarizes the SAMOCP method. It can be observed from (9) that the maximum number of active experts (MOCP instances) at each time $t$ is $g\lfloor \log_2 t \rfloor$. Hence, the complexity of SAMOCP is of order $\mathcal{O}(T \log_2 T)$.

---

**Algorithm 2** Strongly Adaptive Multi-model Ensemble Online Conformal Prediction (SAMOCP)

---

**Require:** $\alpha \in [0, 1]$, hyperparameters $\eta \geq 0, \epsilon \in (0, 1)$, and $\sigma > 1$.
  **for** $t \in [T]$ **do**
    Create new expert $n$ (where $n = t$) by Algorithm 1($\alpha_{t-1}^{\hat{m}\hat{n}}, \eta, \epsilon^n$).
    Remove experts whose lifetime has been finished.
    Every active expert selects miss coverage probability from $M$ options.
    Calculate normalized weights by $\bar{h}_t^n = \frac{h_t^n}{\sum_{i \in \mathcal{A}(t)} h_t^i}$.
    Select one miss overage probability from active experts according to PMF $\bar{h}_t = (\bar{h}_t^n)_{n \in \mathcal{A}(t)}$.
    Construct prediction set for $X_t$ using selected miss coverage probability.
    Observe true label $Y_t^{true}$.
    **for** $n \in \mathcal{A}(t)$ **do**
      Obtain learner loss $L(\bar{\alpha}_t^{\hat{m}\hat{n}}, \alpha_t^{\hat{m}\hat{n}})$.
      Update every parameters assigned to each model for expert $n$ via Algorithm 1.
      Update $h_{t+1}^n$ with (10).
    **end for**
  **end for**

---

Let $CovE(T) := \left| \frac{1}{T} \sum_{t=1}^T \mathbb{E}[err_t] - \alpha \right|$ represent the coverage error. In a dynamic setting where multiple experts are incorporated, each including $M$ miss coverage probabilities, the expected error is calculated as $\mathbb{E}[err_t] = \sum_{n=1}^t \sum_{m=1}^M \bar{h}_t^n \bar{w}_t^{mn} err_t^{mn}$, where $mn$ represents the $m$th model by expert $n$. Using the two following theorems, we prove that SAMOCP has bounded coverage error and achieves strongly adaptive regret across any time interval of arbitrary width (Proofs can be found A.3 and A.4).

**Theorem 2** *For any $T \geq 1$ and any $\gamma \in \left(\frac{1}{2}, 1\right)$, Algorithm 2 achieves the coverage error bound*

$$CovE(T) \leq \mathcal{O}\left(\inf_\gamma \left\{ T^{\frac{1}{2} - \gamma} + T^{\gamma - 1} \beta_\gamma(T) \right\}\right), \tag{11}$$

*where $\beta_\gamma(T)$ measures the smoothness of model weights within experts and the cumulative gradient norm for each model within experts. The definition of $\beta_\gamma(T)$ is provided in detail in equation (31) in the Appendix A.3. If there exists a $\gamma \in \left(\frac{1}{2}, 1\right)$ such that $\beta_\gamma(T) \leq \tilde{\mathcal{O}}(T^\theta)$ where $\theta < 1 - \gamma$, then the coverage bound (11) will be $CovE(T) \leq \tilde{\mathcal{O}}\left(T^{-\min\left(\frac{1}{2} - \gamma, \gamma - 1 + \theta\right)}\right) = \mathbf{o}_T(1)$.*

**Theorem 3** *Algorithm 2 achieves strongly adaptive regret over any interval $I \subseteq [T]$ and positive constants A, B, as follows*

$$\sum_{t \in I} \sum_{n \in \mathcal{A}(t)} \sum_{m=1}^M \bar{h}_t^n \bar{w}_t^{mn} L(\bar{\alpha}_t^{mn}, \alpha_t^{mn}) - \sum_{t \in I} L(\bar{\alpha}_t^{m^*n^*}, \alpha^{m^*n^*}) \leq A\sqrt{|I|} + B \ln T \sqrt{|I|}, \tag{12}$$

*where*

$$\alpha^{m^*n^*} = \arg\min_{\alpha^{m^*n}} \sum_{t \in I} L(\bar{\alpha}_t^{m^*n}, \alpha^{m^*n}).$$ (13)

*Note that in equation* (13)*,* $\alpha^{m^*n}$ *represents the miss coverage probability assigned to the best model for expert* $n$*, as obtained by equation* (7)*.*

The miss coverage probability $\alpha^{m^*n^*}$ in (13), is related to specific interval $I$, which can vary across different intervals with distinct distributions in a dynamic environment. In such settings, there is no fixed miss coverage probability $\alpha^{m^*n^*}$ that can be consistently applied over various time intervals. This necessitates establishing that SAMOCP has bounded regret in dynamic environments with respect to the time-varying benchmark in (16), as demonstrated by the following lemma. The proof is provided in A.5.

**Lemma 1** *By defining the variation of the loss function to be*

$$V(L(\cdot)_{t=1}^T) := \sum_{t=1}^T \max_{\{m \in [M], n \in \mathcal{A}(t)\}} \left| L(\bar{\alpha}_{t+1}^{mn}, \alpha_{t+1}^{mn}) - L(\bar{\alpha}_t^{mn}, \alpha_t^{mn}) \right|.$$ (14)

*We establish the following bound for the dynamic regret of Algorithm 2*

$$\sum_{t=1}^T \sum_{n \in \mathcal{A}(t)} \sum_{m=1}^M \bar{h}_t^n \bar{w}_t^{mn} L(\bar{\alpha}_t^{mn}, \alpha_t^{mn}) - \sum_{t=1}^T L(\bar{\alpha}_t^{m^*n^*}, \alpha_t^{m^*n^*}) \leq \tilde{\mathcal{O}}(T^{\frac{2}{3}} V^{\frac{1}{3}}(L(.)_{t=1}^T))$$ (15)

*where* $\tilde{\mathcal{O}}$ *suppresses positive constants and polylogarithmic factors, e.g.,* $\log T$*. Also the best miss coverage probability at each time* $t$ *can be obtained by*

$$\alpha_t^{m^*n^*} = \arg\min_{\{m \in [M], n \in \mathcal{A}(t)\}} L(\bar{\alpha}_t^{mn}, \alpha_t^{mn}).$$ (16)

Lemma 1 establishes that the dynamic regret of SAMOCP (15) depends on the variation of the loss functions (14). In addition, it can be obtained from (15) that SAMCOP achieves sublinear regret if the variation of the loss function is also sublinear, i.e., $V(L(.)_{t=1}^T) = \mathbf{o}(T)$.

## 4    Experiments

In this section, the performance of the proposed method, SAMOCP, is assessed within the context of classification tasks. We conduct a comprehensive comparison with recently proposed methods in online conformal prediction for dynamic environments within classification tasks. The section begins with a detailed explanation of the experimental settings, followed by a discussion of the results. Note that throughout the experiments in this section, the desired miss coverage probability $\alpha$ is 0.1. All experiments were performed on a workstation with NVIDIA RTX A4000 GPU. Codes are available at hyperrefhttps://github.com/erfanhajihashemi/Multi-model-Ensemble-Conformal-Prediction-in-Dynamic-Environments.

**Dataset:** We utilize corrupted versions of CIFAR-10 and CIFAR-100 [Krizhevsky, 2009], known as CIFAR-10C and CIFAR-100C [Hendrycks and Dietterich, 2019]. These datasets consist of 15 generated corruptions spanning 5 distinct levels of severity. The evaluation encompasses two settings: sudden and gradual distribution shifts. For both settings, the data sequence is split into batches of 500 data samples each. The severity of corruption changes (increases or decreases) after each batch of data. In the sudden shifts, the severity level alternates between the version of the data without any corruption (severity level 0) and the most severe corruption (severity level 5). In the gradual setting, severity starts at level 0 and increases one by one after each batch until it reaches level 5. After reaching level 5, the severity decreases one by one and goes back to level 0 in subsequent batches. This cycle of increasing and decreasing severity continues throughout the experiment. Also, additional experiments on TinyImageNet-C [Hendrycks and Dietterich, 2019] and synthetic data are provided in the Appendix, Section B.

**Baselines and experimental settings:** We employ ResNet-50, ResNet-18 [He et al., 2016], GoogLeNet [Szegedy et al., 2015], and DenseNet-121 [Huang et al., 2017] as candidate learning models. Each active expert consists of all these learning models and needs to select the appropriate model

during its active interval. The proposed method is compared with the most recent adaptive conformal prediction algorithms designed for dynamic environments, including FACI [Gibbs and Candès, 2022], ScaleFreeOGD [Bhatnagar et al., 2023], and SAOCP [Bhatnagar et al., 2023]. FACI employs a fixed number of active experts over all time steps, with each expert assigned one of the candidate learning rates for updating the miss coverage probability. ScaleFreeOGD reduces the learning rate based on the cumulative norms of gradients [Orabona and Pál, 2018]. SAOCP allows each expert to have its own active interval, within which it operates similarly to ScaleFreeOGD. In order to show how SAMOCP results in more efficient sets compared to SAOCP in a multi-model setting, we developed a multi-model ensemble version of SAOCP, denoted as SAOCP(MM), where each expert consists of $M$ update rules, each corresponding to a different learning model. This approach follows our multi-model approach but employs a similar rule to SAOCP for updating weights. To determine the value of $g$, we employed a grid search approach within the candidates $\{4, 8, 16, 24, 32, 48, 64\}$. The one that led to the smallest prediction set size (Avg Width) while maintaining reasonable coverage and runtime was selected, which was $g = 8$. While the hyperparameter $g$ is set to 8 for both SAMOCP and SAOCP(MM), it is set to 32 for SAOCP, as in [Bhatnagar et al., 2023]. Since 4 learning models are incorporated in this section, the maximum number of updates at each time $t$ in SAMOCP, $Mg\lfloor \log_2 t \rfloor$, is equal to that in SAOCP and SAOCP(MM), which is $32\lfloor \log_2 t \rfloor$. Meanwhile, note that randomness might be undesirable in practice, the predicted miss coverage in SAMOCP is calculated in a deterministic fashion, i.e., $\alpha_t = \sum_{n \in \mathcal{A}(t)} \sum_{m=1}^{M} \bar{h}_t^n \bar{w}_t^{mn} \alpha_t^{mn}$. For every experiment conducted on the synthetic dataset, CIFAR-10C, CIFAR-100C, parameters $\epsilon$, $\sigma$, and $\eta$ were selected through grid search, with values of 0.9, 140, and 0.05, respectively.

**Score Functions:** We utilized the nonconformity score defined as in [Angelopoulos et al., 2020] to construct prediction sets. Let

$$S^m(X, Y) = \xi \sqrt{\max([k_Y - k_{reg}], 0)} + U_t \hat{f}_Y^m(X) + \rho(X, Y), \qquad (17)$$

where $\hat{f}_Y^m(X)$ denotes the probability of predicting label $Y$ for input $X$ by model $m$, and $U_t$ is a random variable sampled from a uniform distribution over the interval $[0, 1]$. The term $k_Y :=$ $|\{Y' \in \mathcal{Y} \mid \hat{f}_{Y'}^m(X) \geq \hat{f}_Y^m(X)\}|$ denotes the number of labels that have a higher or equal predicted probability than label $Y$ according to the model's output probability distribution, e.g., the softmax output. $\rho(X, Y) := \sum_{Y'=1}^{K} \hat{f}_{Y'}^m(X) \mathbb{I}[\hat{f}_{Y'}^m(X) > \hat{f}_Y^m(X)]$ sums up the probabilities of all labels that have a higher predicted probability than label $Y$. The hyperparameters $\xi$ and $k_{reg}$ are set to 0.02 and 5 for CIFAR-100C, and 0.1 and 1 for Cifar-10C, respectively.

**Evaluation Metrics:** Coverage measures the percentage of instances where the true label is included in the prediction sets outputted by the conformal prediction algorithm over the period $[T]$. Avg Width refers to the average size of these prediction sets. Adaptive regret is calculated for time intervals of length 100. The metric Avg Regret represents the average of these regret values across the entire time horizon $[T]$. Run Time indicates the time required to complete each iteration of the algorithm. Lastly, Single Width measures the probability that prediction sets contain exactly one element while accurately covering the true label, highlighting cases that are most informative for predictions.

## 4.1 Results

Table 1 presents the performance of SAMOCP for the classification task on the CIFAR-100C dataset under a gradual shift setting, where each method receives $8, 550$ data points sequentially. It is evident that the performance of previous methods, particularly in terms of prediction set size and single-width prediction sets, depends on the learning model employed. The proposed method SAMOCP outperforms existing methods by creating smaller prediction sets, lower regret, and more single-width prediction sets that correctly cover the true label, while also achieving coverage close to the targeted level. It is noteworthy that SAMOCP surpasses every variant of previous methods in these aspects. Furthermore, SAMOCP is faster than SAOCP and SAOCP(MM), despite having the same maximum number of updates at each time $t$.

In dynamic environments, data distribution does not necessarily shift gradually. Instead, we may encounter abrupt distribution shifts, with significant differences between data distributions in two successive time slots. To demonstrate how SAMOCP behaves in such environments, another experiment was conducted, in which SAMOCP can successfully track these sharp transitions and select a suitable learning model for creating a prediction set. Experimental results on CIFAR-10C are detailed in

Table 1: Results on the CIFAR-100C dataset with a gradual distribution shift. The target coverage is 90%, and the average regret is calculated over an interval size of 100. Bold numbers denote the best results in each column. SAMOCP achieves the best performance in terms of average width, average regret, and single width.

| Model | Method | Coverage (%) | Avg Width | Avg Regret($\times 10^{-3}$) | Run Time | Single Width |
|---|---|---|---|---|---|---|
| | SAMOCP | $88.16 \pm 0.18$ | $\mathbf{5.43 \pm 0.28}$ | $\mathbf{0.92 \pm 0.07}$ | $34.87 \pm 0.67$ | $\mathbf{0.29 \pm 0.01}$ |
| | SAOCP(MM) | $85.89 \pm 0.84$ | $6.61 \pm 0.26$ | $4.42 \pm 0.51$ | $47.80 \pm 0.22$ | $0.27 \pm 0.01$ |
| DenseNet-121 | FACI | $89.64 \pm 0.28$ | $5.77 \pm 0.62$ | $1.18 \pm 0.74$ | $8.22 \pm 0.07$ | $0.28 \pm 0.01$ |
| | ScaleFreeOGD | $\mathbf{89.97 \pm 0.02}$ | $6.04 \pm 0.10$ | $1.55 \pm 0.06$ | $\mathbf{3.02 \pm 0.06}$ | $0.26 \pm 0.01$ |
| | SAOCP | $88.90 \pm 0.10$ | $5.80 \pm 0.08$ | $2.04 \pm 0.04$ | $40.74 \pm 0.31$ | $0.27 \pm 0.00$ |
| ResNet-50 | FACI | $89.67 \pm 0.33$ | $6.50 \pm 0.57$ | $1.24 \pm 0.90$ | $8.45 \pm 0.09$ | $0.26 \pm 0.01$ |
| | ScaleFreeOGD | $\mathbf{89.97 \pm 0.02}$ | $6.72 \pm 0.13$ | $1.58 \pm 0.07$ | $3.12 \pm 0.04$ | $0.25 \pm 0.01$ |
| | SAOCP | $88.79 \pm 0.14$ | $6.48 \pm 0.13$ | $2.08 \pm 0.10$ | $41.35 \pm 0.34$ | $0.25 \pm 0.00$ |
| ResNet-18 | FACI | $89.56 \pm 0.28$ | $6.82 \pm 0.77$ | $1.17 \pm 0.75$ | $8.39 \pm 0.06$ | $0.25 \pm 0.01$ |
| | ScaleFreeOGD | $89.96 \pm 0.02$ | $7.29 \pm 0.19$ | $1.55 \pm 0.07$ | $3.05 \pm 0.04$ | $0.23 \pm 0.01$ |
| | SAOCP | $88.76 \pm 0.22$ | $6.9 \pm 0.17$ | $2.06 \pm 0.07$ | $41.23 \pm 0.26$ | $0.24 \pm 0.01$ |
| GoogLeNet | FACI | $89.63 \pm 0.30$ | $6.33 \pm 0.74$ | $1.10 \pm 0.78$ | $8.30 \pm 0.09$ | $0.27 \pm 0.01$ |
| | ScaleFreeOGD | $89.96 \pm 0.01$ | $6.71 \pm 0.15$ | $1.52 \pm 0.06$ | $3.04 \pm 0.04$ | $0.24 \pm 0.00$ |
| | SAOCP | $88.68 \pm 0.11$ | $6.38 \pm 0.12$ | $2.07 \pm 0.07$ | $41.13 \pm 0.39$ | $0.26 \pm 0.01$ |

Table 2, where the proposed algorithm again outperforms previous methods in terms of prediction set size, regret, and single width prediction sets that accurately cover the true labels while maintaining coverage close to the target value.

To demonstrate that SAMOCP achieves the lowest regret over different intervals compared to existing methods, we illustrate the regret for various interval sizes in Figure 2. For each existing method, there are 4 different regrets corresponding to the 4 learning models used and the lowest regret is depicted. The results show that our method consistently leads to lower regret than the best version of each previous method across different learning models. Note that a lower regret implies that the algorithm adapts faster to changes. The regret calculated over different time intervals indicates the algorithm's adaptivity in capturing the distribution shift at different time scales. Therefore, Figure 2 indicates that the SAMOCP can adapt faster to distribution shifts compared to benchmarks in various time scales. Furthermore, we include experiments on synthetic data and another real dataset, TinyImageNet-C, using different sets of learning models that do not necessarily contain 4 models in the Appendix, Section B. This demonstrates how SAMOCP can rely on a mixture of learning models over the period $[T]$ and select the appropriate one for each distribution setting.

Table 2: Results on the CIFAR-10C dataset with a sudden distribution shift. The target coverage is 90%, and the average regret is calculated over an interval size of 100. Bold numbers denote the best results in each column. SAMOCP achieves the best performance in terms of average width, average regret, and single width.

| Model | Method | Coverage (%) | Avg Width | Avg Regret($\times 10^{-3}$) | Run Time | Single Width |
|---|---|---|---|---|---|---|
| | SAMOCP | $88.37 \pm 0.23$ | $\mathbf{1.24 \pm 0.06}$ | $\mathbf{0.98 \pm 0.11}$ | $33.75 \pm 0.34$ | $\mathbf{0.69 \pm 0.03}$ |
| | SAOCP(MM) | $86.80 \pm 2.39$ | $1.45 \pm 0.13$ | $3.87 \pm 1.05$ | $47.08 \pm 0.19$ | $0.56 \pm 0.05$ |
| DenseNet-121 | FACI | $89.57 \pm 0.37$ | $1.30 \pm 0.12$ | $1.46 \pm 0.73$ | $8.11 \pm 0.10$ | $0.68 \pm 0.05$ |
| | ScaleFreeOGD | $\mathbf{89.99 \pm 0.01}$ | $1.46 \pm 0.02$ | $1.71 \pm 0.04$ | $2.92 \pm 0.07$ | $0.52 \pm 0.02$ |
| | SAOCP | $88.77 \pm 0.18$ | $1.41 \pm 0.02$ | $2.24 \pm 0.06$ | $39.62 \pm 0.22$ | $0.54 \pm 0.01$ |
| ResNet-50 | FACI | $89.74 \pm 0.35$ | $1.50 \pm 0.04$ | $1.35 \pm 0.93$ | $8.11 \pm 0.08$ | $0.55 \pm 0.01$ |
| | ScaleFreeOGD | $89.98 \pm 0.01$ | $1.52 \pm 0.01$ | $1.71 \pm 0.05$ | $\mathbf{2.89 \pm 0.04}$ | $0.54 \pm 0.01$ |
| | SAOCP | $89.12 \pm 0.08$ | $1.51 \pm 0.01$ | $2.17 \pm 0.06$ | $40.25 \pm 0.27$ | $0.53 \pm 0.01$ |
| ResNet-18 | FACI | $89.63 \pm 0.34$ | $1.36 \pm 0.13$ | $1.52 \pm 0.76$ | $8.11 \pm 0.08$ | $0.64 \pm 0.05$ |
| | ScaleFreeOGD | $\mathbf{89.99 \pm 0.01}$ | $1.52 \pm 0.02$ | $1.69 \pm 0.06$ | $2.91 \pm 0.06$ | $0.49 \pm 0.01$ |
| | SAOCP | $88.83 \pm 0.06$ | $1.48 \pm 0.02$ | $2.24 \pm 0.07$ | $40.16 \pm 0.22$ | $0.51 \pm 0.01$ |
| GoogLeNet | FACI | $89.73 \pm 0.34$ | $1.43 \pm 0.06$ | $1.41 \pm 0.89$ | $8.10 \pm 0.10$ | $0.58 \pm 0.02$ |
| | ScaleFreeOGD | $89.99 \pm 0.02$ | $1.46 \pm 0.01$ | $1.70 \pm 0.06$ | $2.89 \pm 0.04$ | $0.55 \pm 0.00$ |
| | SAOCP | $89.09 \pm 0.14$ | $1.44 \pm 0.01$ | $2.17 \pm 0.08$ | $40.13 \pm 0.17$ | $0.55 \pm 0.01$ |

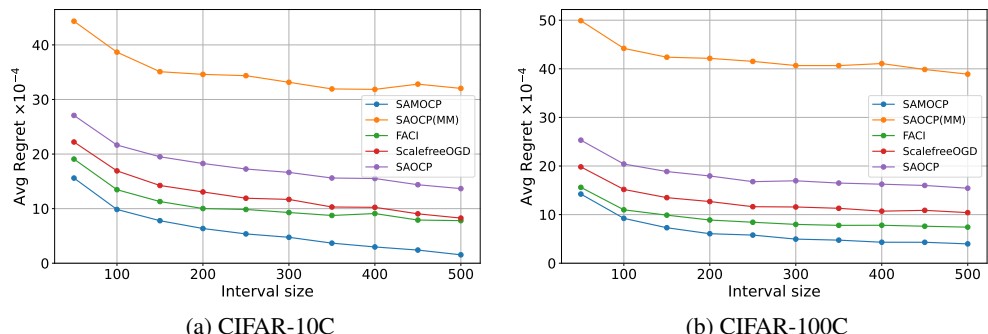

(a) CIFAR-10C               (b) CIFAR-100C

Figure 2: Evaluation of average regret over different interval sizes $(50, 100, \ldots, 500)$. Note that for previous methods relying on a single model, the lowest regret across the $4$ learning models is selected.

## Acknowledgement

Work in this paper is supported by NSF ECCS 2207457 and NSF ECCS 2412484.

## Conclusion

In this study, we introduced a novel conformal prediction algorithm designed for online environments undergoing distribution shifts. Recognizing that the selection of baseline models affects the efficiency of conformal prediction, our algorithm incorporates multiple models simultaneously. For each expert, the contribution of each model is dynamically adjusted based on its time-evolving weight. We demonstrated that our proposed method SAMOCP achieves strongly adaptive regret across any time interval of arbitrary width and maintains valid coverage. Experimental results in environments with both gradual and sudden distribution shifts indicated that our algorithm produces more informative prediction sets and achieves a coverage rate close to the target value, compared to those created by previous methods using their best baseline models.

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

# A Proofs

**Lemma 2** *For every $\alpha \in [0, 1]$ and learning rate $\eta > 0$, adaptive miss coverage probability $\alpha_t^m$ for any model $m \in [M]$ and $t \in [T]$ is bounded as*

$$\alpha_t^m \in [-\eta, 1 + \eta].$$

## A.1 Proof of lemma 2

Based on equation (4), we have

$$|\alpha_t^m - \alpha_{t-1}^m| = \eta \left| \frac{(\alpha - err_{t-1}^m)}{\sqrt{\sum_{\tau=1}^{t-1} \|\nabla_{\alpha_\tau^m} L(\bar{\alpha}_\tau^m, \alpha_\tau^m)\|_2^2}} \right| \leq \eta. \tag{18}$$

We prove this lemma using a contradiction. Suppose there exists a $\hat{t}$ such that $\alpha_{\hat{t}}^m \notin [-\eta, 1 + \eta]$, where $\hat{t} \geq 2$ is the smallest such time index. We first prove the case of violating the upper bound; by contradiction, assume $\alpha_{\hat{t}}^m > 1 + \eta$. According to equation (18), this would necessitate that $\alpha_{\hat{t}-1}^m \geq 1$. Given that we assumed $\hat{t}$ is the smallest time index to violate the upper bound, it should follow that $\alpha_{\hat{t}-1}^m \leq 1 + \eta$. However, $\alpha_{\hat{t}-1}^m > 1 > \bar{\alpha}_{\hat{t}-1}^m$ implies $err_{\hat{t}-1}^m = 1$. By equation (4) we have

$$\alpha_{\hat{t}}^m = \alpha_{\hat{t}-1}^m + \eta \frac{\alpha - 1}{\sqrt{\sum_{\tau=1}^{\hat{t}-1} \|\nabla_{\alpha_\tau^m} L(\bar{\alpha}_\tau^m, \alpha_\tau^m)\|_2^2}} \leq \alpha_{\hat{t}-1}^m \leq 1 + \eta,$$

which contradicts our assumption that $\alpha_{\hat{t}-1}^m > 1 + \eta$. Next, assume $\alpha_{\hat{t}}^m < -\eta$. By equation (18), we have $\alpha_{\hat{t}-1}^m < 0$. Given that $\hat{t}$ is the smallest index that violates the lower bound of the lemma, it must hold that $\alpha_{\hat{t}-1}^m \geq -\eta$. Considering $\alpha_{\hat{t}-1}^m < 0 < \bar{\alpha}_{\hat{t}-1}^m$, we deduce that $err_{\hat{t}-1}^m = 0$. Therefore, by equation (4), we have

$$\alpha_{\hat{t}}^m = \alpha_{\hat{t}-1}^m + \eta \frac{\alpha}{\sqrt{\sum_{\tau=1}^{\hat{t}-1} \|\nabla_{\alpha_\tau^m} L(\bar{\alpha}_\tau^m, \alpha_\tau^m)\|_2^2}} \geq \alpha_{\hat{t}-1}^m \geq -\eta,$$

which contradicts our initial assumption that $\alpha_{\hat{t}}^m < -\eta$.

## A.2 Proof of Theorem 1, Regret for MOCP:

Algorithm 1 has the following regret bound

$$\sum_{t=1}^{T} \sum_{m=1}^{M} \bar{w}_t^m L(\bar{\alpha}_t^m, \alpha_t^m) - \sum_{t=1}^{T} L(\bar{\alpha}_t^{m^*}, \alpha^{m^*}) \leq \sqrt{T} \left( \frac{(1 + 2\eta)^2}{2\eta} + \frac{\eta}{2\alpha} + \ln M + (1 + \eta)^2 \right),$$

Where $\alpha^{m^*}$ can be obtained via (7). To prove the theorem, we first introduce and prove the following two lemmas

**Lemma 3** *for miss coverage probability assigned to any model $\tilde{m} \in [M]$, we have the following bound*

$$\sum_{t=1}^{T} L(\bar{\alpha}_t^{\tilde{m}}, \alpha_t^{\tilde{m}}) - \sum_{t=1}^{T} L(\bar{\alpha}_t^{\tilde{m}}, \alpha^{\tilde{m}}) \leq \frac{\sqrt{T}}{2\eta} (1 + 2\eta)^2 + \frac{\eta \sqrt{T}}{2\alpha},$$

*where $\alpha^{\tilde{m}} = \arg \min_{\alpha^{\tilde{m}}} \sum_{t=1}^{T} L(\bar{\alpha}_t^{\tilde{m}}, \alpha_t^{\tilde{m}})$.*
***Proof:*** *We first begin with*

$$(\alpha_{t+1}^{\tilde{m}} - \alpha^{\tilde{m}})^2 = (\alpha_t^{\tilde{m}} - \eta \frac{\nabla_{\alpha_t^{\tilde{m}}} L(\bar{\alpha}_t^{\tilde{m}}, \alpha_t^{\tilde{m}})}{\sqrt{\sum_{\tau=1}^{t} \|\nabla_{\alpha_\tau^{\tilde{m}}} L(\bar{\alpha}_\tau^{\tilde{m}}, \alpha_\tau^{\tilde{m}})\|_2^2}} - \alpha^{\tilde{m}})^2.$$

*Then define adaptive learning rate $\eta_t$ [Duchi et al., 2011, Hazan et al., 2007] as*

$$\eta_t := \frac{\eta}{\sqrt{\sum_{\tau=1}^{t} \|\nabla_{\alpha_\tau^{\tilde{m}}} L(\bar{\alpha}_\tau^{\tilde{m}}, \alpha_\tau^{\tilde{m}})\|_2^2}}.$$

*So we have*

$$(\alpha_{t+1}^{\tilde{m}} - \alpha^{\tilde{m}})^2 = (\eta_t \nabla_{\alpha_t^{\tilde{m}}} L(\tilde{\alpha}_t^{\tilde{m}}, \alpha_t^{\tilde{m}}))^2 + (\alpha_t^{\tilde{m}} - \alpha^{\tilde{m}})^2 - 2\eta_t(\alpha_t^{\tilde{m}} - \alpha^{\tilde{m}})\nabla_{\alpha_t^{\tilde{m}}} L(\tilde{\alpha}_t^{\tilde{m}}, \alpha_t^{\tilde{m}}).$$

*Therefore,*

$$(\alpha_t^{\tilde{m}} - \alpha^{\tilde{m}})\nabla_{\alpha_t^{\tilde{m}}} L(\bar{\alpha}_t^{\tilde{m}}, \alpha_t^{\tilde{m}}) = \frac{(\alpha_t^{\tilde{m}} - \alpha^{\tilde{m}})^2 - (\alpha_{t+1}^{\tilde{m}} - \alpha^{\tilde{m}})^2}{2\eta_t} + \frac{\eta_t}{2}(\nabla_{\alpha_t^{\tilde{m}}} L(\bar{\alpha}_t^{\tilde{m}}, \alpha_t^{\tilde{m}}))^2.$$

*Since the loss function* (2) *is convex, we have the following inequality*

$$L(\bar{\alpha}_t^{\tilde{m}}, \alpha_t^{\tilde{m}}) - L(\bar{\alpha}_t^{\tilde{m}}, \alpha^{\tilde{m}}) \leq (\alpha_t^{\tilde{m}} - \alpha^{\tilde{m}})\nabla_{\alpha_t^{\tilde{m}}} L(\bar{\alpha}_t^{\tilde{m}}, \alpha_t^{\tilde{m}}). \tag{19}$$

*By summing* (19) *over* $t \in [T]$ *we have*

$$\sum_{t=1}^{T} \left( L\left(\bar{\alpha}_t^{\tilde{m}}, \alpha_t^{\tilde{m}}\right) - L\left(\bar{\alpha}_t^{\tilde{m}}, \alpha^{\tilde{m}}\right) \right)$$

$$\leq \sum_{t=1}^{T} \frac{(\alpha_t^{\tilde{m}} - \alpha^{\tilde{m}})^2 - (\alpha_{t+1}^{\tilde{m}} - \alpha^{\tilde{m}})^2}{2\eta_t} + \sum_{t=1}^{T} \frac{\eta_t}{2}\left( \nabla_{\alpha_t^{\tilde{m}}} L(\bar{\alpha}_t^{\tilde{m}}, \alpha_t^{\tilde{m}}) \right)^2$$

$$\leq \frac{\sqrt{T}}{2\eta} \sum_{t=1}^{T} \left( (\alpha_t^{\tilde{m}} - \alpha^{\tilde{m}})^2 - (\alpha_{t+1}^{\tilde{m}} - \alpha^{\tilde{m}})^2 \right) + \frac{\eta}{2} \sum_{t=1}^{T} \frac{1}{\sqrt{\sum_{\tau=1}^{t} \|\nabla_{\alpha_\tau^{\tilde{m}}} L(\bar{\alpha}_\tau^{\tilde{m}}, \alpha_\tau^{\tilde{m}})\|_2^2}}$$

$$\leq \frac{\sqrt{T}}{2\eta} \left( (\alpha_1^{\tilde{m}} - \alpha^{\tilde{m}})^2 - (\alpha_{T+1}^{\tilde{m}} - \alpha^{\tilde{m}})^2 \right) + \frac{\eta}{2} \sum_{t=1}^{T} \frac{1}{\alpha\sqrt{T}} \overset{(i)}{\leq} \frac{\sqrt{T}}{2\eta}(1 + 2\eta)^2 + \frac{\eta\sqrt{T}}{2\alpha}, \tag{20}$$

*where* (i) *used* $\alpha_t^m \in [-\eta, 1 + \eta]$ *by Lemma 2.*

**Lemma 4** *For miss coverage probability assigned to any model* $\tilde{m} \in [M]$ *we have the following bound*

$$\sum_{t=1}^{T} \sum_{m=1}^{M} \bar{w}_t^m L(\bar{\alpha}_t^m, \alpha_t^m) - \sum_{t=1}^{T} L(\bar{\alpha}_t^{\tilde{m}}, \alpha_t^{\tilde{m}}) \leq \frac{\ln M}{\epsilon} + \epsilon(1 + \eta)^2 T.$$

**Proof:** *Referring to the definition of* $\bar{w}_t^m$ *in Subsection 3.1 and defining* $W_t := \sum_{m=1}^{M} w_t^m$, *we have*

$$W_{T+1} = \sum_{m=1}^{M} w_{T+1}^m = \sum_{m=1}^{M} w_T^m \exp\left(-\epsilon L(\bar{\alpha}_T^m, \alpha_T^m)\right) = W_T \sum_{m=1}^{M} \bar{w}_T^m \exp\left(-\epsilon L(\bar{\alpha}_T^m, \alpha_T^m)\right)$$

$$\overset{(i)}{\leq} W_T \sum_{m=1}^{M} \bar{w}_T^m \left(1 - \epsilon L(\bar{\alpha}_T^m, \alpha_T^m) + \epsilon^2 L(\bar{\alpha}_T^m, \alpha_T^m)^2\right)$$

$$= W_T \left(1 - \epsilon \sum_{m=1}^{M} \bar{w}_T^m L(\bar{\alpha}_T^m, \alpha_T^m) + \epsilon^2 \sum_{m=1}^{M} \bar{w}_T^m L(\bar{\alpha}_T^m, \alpha_T^m)^2\right)$$

$$\overset{(ii)}{\leq} W_T \exp\left(-\epsilon \sum_{m=1}^{M} \bar{w}_T^m L(\bar{\alpha}_T^m, \alpha_T^m) + \epsilon^2 \sum_{m=1}^{M} \bar{w}_T^m L(\bar{\alpha}_T^m, \alpha_T^m)^2\right)$$

$$\leq W_1 \exp\left(-\epsilon \sum_{t=1}^{T} \sum_{m=1}^{M} \bar{w}_t^m L(\bar{\alpha}_t^m, \alpha_t^m) + \epsilon^2 \sum_{t=1}^{T} \sum_{m=1}^{M} \bar{w}_t^m L(\bar{\alpha}_t^m, \alpha_t^m)^2\right), \tag{21}$$

*where* $W_1 = 1$, (i) *follows from the inequality* $\exp(-\epsilon x) \leq 1 - \epsilon x + \epsilon^2 x^2$ *for* $|\epsilon| \leq 1$, *and* (ii) *follows from* $1 + x \leq e^x$. *On the other hand, we have*

$$W_{T+1} \geq w_{T+1}^{\tilde{m}} = w_1^{\tilde{m}} \prod_{t=1}^{T} \exp\left(-\epsilon L(\bar{\alpha}_t^{\tilde{m}}, \alpha_t^{\tilde{m}})\right) = w_1^{\tilde{m}} \exp\left(-\epsilon \sum_{t=1}^{T} L(\bar{\alpha}_t^{\tilde{m}}, \alpha_t^{\tilde{m}})\right), \tag{22}$$

*where $w_1^{\bar{m}} = \frac{1}{M}$. By combining (21) and (22) we have*

$$\exp\left(-\epsilon \sum_{t=1}^{T}\sum_{m=1}^{M} \bar{w}_t^m L(\bar{\alpha}_t^m, \alpha_t^m) + \epsilon^2 \sum_{t=1}^{T}\sum_{m=1}^{M} \bar{w}_t^m L(\bar{\alpha}_t^m, \alpha_t^m)^2\right)$$

$$\geq \frac{1}{M}\exp\left(-\epsilon \sum_{t=1}^{T} L(\bar{\alpha}_t^{\tilde{m}}, \alpha_t^{\tilde{m}})\right). \tag{23}$$

*By taking the logarithm on both sides we have*

$$-\epsilon \sum_{t=1}^{T}\sum_{m=1}^{M} \bar{w}_t^m L(\bar{\alpha}_t^m, \alpha_t^m) + \epsilon^2 \sum_{t=1}^{T}\sum_{m=1}^{M} \bar{w}_t^m L(\bar{\alpha}_t^m, \alpha_t^m)^2 \geq -\ln M - \epsilon \sum_{t=1}^{T} L(\bar{\alpha}_t^{\tilde{m}}, \alpha_t^{\tilde{m}}),$$

*which leads to*

$$\sum_{t=1}^{T}\sum_{m=1}^{M} \bar{w}_t^m L(\bar{\alpha}_t^m, \alpha_t^m) - \sum_{t=1}^{T} L(\bar{\alpha}_t^{\tilde{m}}, \alpha_t^{\tilde{m}}) \leq \frac{\ln M}{\epsilon} + T\epsilon(1+\eta)^2. \tag{24}$$

Now, we define the best model in the static environment as

$$m^* = \arg\min_{m \in M} \sum_{t=1}^{T} L(\bar{\alpha}_t^m, \alpha_t^m).$$

Then, we replace $\tilde{m}$ with best model $m^*$ in Lemma 3 and 4. By setting $\epsilon = \frac{1}{\sqrt{T}}$ and summing results of two lemmas we have:

$$\sum_{t=1}^{T}\sum_{m=1}^{M} \bar{w}_T^m L(\bar{\alpha}_t^m, \alpha_t^m) - \sum_{t=1}^{T} L(\bar{\alpha}_t^{m^*}, \alpha_t^{m^*}) + \sum_{t=1}^{T} L(\bar{\alpha}_t^{m^*}, \alpha_t^{m^*}) - \sum_{t=1}^{T} L(\bar{\alpha}_t^{m^*}, \alpha^{m^*})$$

$$= \sum_{t=1}^{T}\sum_{m=1}^{M} \bar{w}_T^m L(\bar{\alpha}_t^m, \alpha_t^m) - \sum_{t=1}^{T} L(\bar{\alpha}_t^{m^*}, \alpha^{m^*}) \leq \sqrt{T}\left(\frac{(1+2\eta)^2}{2\eta} + \frac{\eta}{2\alpha} + \ln M + (1+\eta)^2\right). \tag{25}$$

### A.3 Proof of Theorem 2, Coverage error for SAMOCP:

We first define expected miss coverage error as

$$\mathbb{E}[err_t] = \sum_{n=1}^{t}\sum_{m=1}^{M} \bar{h}_t^n \bar{w}_t^{mn} err_t^{mn}.$$

The proof of this theorem is based on a grouping argument. So we first divide $T$ into $\lceil T^{1-\gamma}\rceil$ group for $\gamma \in (\frac{1}{2}, 1)$, and write the $k$th group as

$$G_k = \{t_{k-1}+1, \ldots, \min(t_k, T)\}.$$

where $|G_k| \leq \lceil T^\gamma\rceil$. We also define a new variable, $H_{n:t}^{mn}$, assigned to $m$th update rule of $n$th expert as follows

$$H_{n:t}^{mn} := \sqrt{\sum_{\tau=n}^{t} \|\nabla_{\alpha_\tau^{mn}} L(\bar{\alpha}_\tau^{mn}, \alpha_\tau^{mn})\|_2^2}.$$

So the update rule in (4) can be written as:

$$\alpha_{t+1}^{mn} = \alpha_t^{mn} + \eta\frac{(\alpha - err_t^{mn})}{H_{n:t}^{mn}}. \tag{26}$$

For $k$th group where $2 \leq k \leq \lceil T^{1-\gamma}\rceil$, by using (26) we have

$$\mathbb{E}[err_t] - \alpha = \sum_{n=1}^{t}\sum_{m=1}^{M} \bar{h}_t^n \bar{w}_t^{mn}(err_t^{mn} - \alpha) = \frac{1}{\eta}\sum_{n=1}^{t}\sum_{m=1}^{M} \bar{h}_t^n \bar{w}_t^{mn}(\alpha_t^{mn} - \alpha_{t+1}^{mn})H_{n:t}^{mn}.$$

Since at each time $t$ we activate an expert with lifetime as defined in (9), the $n$th expert will be activated at time $t = n$. Consequently, $\bar{h}_t^n$ will be 0 for $t < n$. Therefore, we have

$$\frac{1}{\eta}\sum_{n=1}^{t}\sum_{m=1}^{M}\bar{h}_t^n\bar{w}_t^{mn}(\alpha_t^{mn}-\alpha_{t+1}^{mn})H_{n:t}^{mn} = \frac{1}{\eta}\sum_{n=1}^{t_{k-1}}\sum_{m=1}^{M}\bar{h}_{t_{k-1}}^n\bar{w}_{t_{k-1}}^{mn}(\alpha_t^{mn}-\alpha_{t+1}^{mn})H_{n:t_{k-1}}^{mn}$$

$$+\frac{1}{\eta}\sum_{n=1}^{t}\sum_{m=1}^{M}\bar{h}_t^n\bar{w}_t^{mn}(\alpha_t^{mn}-\alpha_{t+1}^{mn})H_{n:t}^{mn}-\frac{1}{\eta}\sum_{n=1}^{t_{k-1}}\sum_{m=1}^{M}\bar{h}_{t_{k-1}}^n\bar{w}_{t_{k-1}}^{mn}(\alpha_t^{mn}-\alpha_{t+1}^{mn})H_{n:t_{k-1}}^{mn}$$

$$=\frac{1}{\eta}\sum_{n=1}^{t_{k-1}}\sum_{m=1}^{M}\bar{h}_{t_{k-1}}^n\bar{w}_{t_{k-1}}^{mn}(\alpha_t^{mn}-\alpha_{t+1}^{mn})H_{n:t_{k-1}}^{mn}$$

$$+\frac{1}{\eta}\sum_{n=1}^{t}\sum_{m=1}^{M}(\bar{h}_t^n\bar{w}_t^{mn}-\bar{h}_{t_{k-1}}^n\bar{w}_{t_{k-1}}^{mn}\frac{H_{n:t_{k-1}}^{mn}}{H_{n:t}^{mn}})(\alpha_t^{mn}-\alpha_{t+1}^{mn})H_{n:t}^{mn}. \qquad (27)$$

Note that according to (26), $H_{n:t}^{mn}(\alpha_t^{mn}-\alpha_{t+1}^{mn})\le\eta$. By summing (27) over $t\in G_k$ we have

$$\left|\sum_{t\in G_k}(\mathbb{E}[err_t]-\alpha)\right|\le\left|\frac{1}{\eta}\sum_{n=1}^{t_{k-1}}\sum_{m=1}^{M}\bar{h}_{t_{k-1}}^n\bar{w}_{t_{k-1}}^{mn}H_{n:t_{k-1}}^{mn}\sum_{t\in G_k}(\alpha_t^{mn}-\alpha_{t+1}^{mn})\right|$$

$$+\left|\frac{1}{\eta}\sum_{t\in G_k}\sum_{n=1}^{t}\sum_{m=1}^{M}(\bar{h}_t^n\bar{w}_t^{mn}-\bar{h}_{t_{k-1}}^n\bar{w}_{t_{k-1}}^{mn}\frac{H_{n:t_{k-1}}^{mn}}{H_{n:t}^{mn}})(\alpha_t^{mn}-\alpha_{t+1}^{mn})H_{n:t}^{mn}\right|$$

$$\le\left|\frac{1}{\eta}\sum_{n=1}^{t_{k-1}}\sum_{m=1}^{M}\bar{h}_{t_{k-1}}^n\bar{w}_{t_{k-1}}^{mn}H_{n:t_{k-1}}^{mn}(\alpha_{t_{k-1}+1}^{mn}-\alpha_{t_k+1}^{mn})\right|$$

$$+\left|\sum_{t\in G_k}\sum_{n=1}^{t}\sum_{m=1}^{M}(\bar{h}_t^n\bar{w}_t^{mn}-\bar{h}_{t_{k-1}}^n\bar{w}_{t_{k-1}}^{mn}\frac{H_{n:t_{k-1}}^{mn}}{H_{n:t}^{mn}})\right|$$

$$\le\frac{1}{\eta}\max_{\{m\in[M],n\in[t_{k-1}]\}}H_{n:t_{k-1}}^{mn}\left|\alpha_{t_{k-1}+1}^{mn}-\alpha_{t_k+1}^{mn}\right|$$

$$+|G_k|\cdot\max_{t\in G_k}\sum_{n=1}^{t}\sum_{m=1}^{M}\left|\bar{h}_t^n\bar{w}_t^{mn}-\bar{h}_{t_{k-1}}^n\bar{w}_{t_{k-1}}^{mn}\frac{H_{n:t_{k-1}}^{mn}}{H_{n:t}^{mn}}\right|$$

$$\le\frac{1+2\eta}{\eta}\sqrt{T}+\lceil T^\gamma\rceil\cdot\max_{t\in G_k}\sum_{n=1}^{t}\sum_{m=1}^{M}\left|\bar{h}_t^n\bar{w}_t^{mn}-\bar{h}_{t_{k-1}}^n\bar{w}_{t_{k-1}}^{mn}\frac{H_{n:t_{k-1}}^{mn}}{H_{n:t}^{mn}}\right| \qquad (28)$$

For $G_1$ we have

$$\left|\sum_{t\in G_1}\mathbb{E}[err_t]-\alpha\right|=\sum_{t\in G_1}\sum_{n=1}^{t}\sum_{m=1}^{M}\bar{h}_t^n\bar{w}_t^{mn}(err_t^{mn}-\alpha)\le\sum_{t\in G_1}\sum_{n=1}^{t}\sum_{m=1}^{M}\bar{h}_t^n\bar{w}_t^{mn}\le|G_1|\le\lceil T^\gamma\rceil.$$

$$(29)$$

By summing over all group we have

$$\left|\sum_{t=1}^{T}\mathbb{E}[err_t]-\alpha\right|=\sum_{k=1}^{\lceil T^{1-\gamma}\rceil}\left|\sum_{t\in G_k}\mathbb{E}[err_t]-\alpha\right|$$

$$\le 2T^\gamma+\sum_{k=2}^{\lceil T^{1-\gamma}\rceil}\left(\frac{1+2\eta}{\eta}\sqrt{T}+2T^\gamma\cdot\max_{t\in G_k}\sum_{n=1}^{t}\sum_{m=1}^{M}\left|\bar{h}_t^n\bar{w}_t^{mn}-\bar{h}_{t_{k-1}}^n\bar{w}_{t_{k-1}}^{mn}\frac{H_{n:t_{k-1}}^{mn}}{H_{n:t}^{mn}}\right|\right)$$

$$\le T^{\frac{3}{2}-\gamma}\frac{1+2\eta}{\eta}+2T^\gamma\left(1+\sum_{k=2}^{\lceil T^{1-\gamma}\rceil}\max_{t\in G_k}\sum_{n=1}^{t}\sum_{m=1}^{M}\left|\bar{h}_t^n\bar{w}_t^{mn}-\bar{h}_{t_{k-1}}^n\bar{w}_{t_{k-1}}^{mn}\frac{H_{n:t_{k-1}}^{mn}}{H_{n:t}^{mn}}\right|\right)$$

$$\le\mathcal{O}(T^{\frac{3}{2}-\gamma}+T^\gamma\left(1+\sum_{k=2}^{\lceil T^{1-\gamma}\rceil}\max_{t\in G_k}\sum_{n=1}^{t}\sum_{m=1}^{M}\left|\bar{h}_t^n\bar{w}_t^{mn}-\bar{h}_{t_{k-1}}^n\bar{w}_{t_{k-1}}^{mn}\frac{H_{n:t_{k-1}}^{mn}}{H_{n:t}^{mn}}\right|\right) \qquad (30)$$

We define $\beta_\gamma(T)$ as:

$$\beta_\gamma(T) := \left(1 + \sum_{k=2}^{\lceil T^{1-\gamma} \rceil} \max_{t \in G_k} \sum_{n=1}^{t} \sum_{m=1}^{M} \left| \bar{h}_t^n \bar{w}_t^{mn} - \bar{h}_{t_{k-1}}^n \bar{w}_{t_{k-1}}^{mn} \frac{H_{n:t_{k-1}}^{mn}}{H_{n:t}^{mn}} \right| \right). \tag{31}$$

Then we have:

$$\left| \sum_{t=1}^{T} \mathbb{E}[err_t] - \alpha \right| = \mathcal{O}(T^{\frac{3}{2}-\gamma} + T^\gamma \beta_\gamma(T)).$$

## A.4 Proof of Theorem 3, static regret for SAMOCP

We can write the regret as

$$\sum_{t \in I_{\tilde{n}}} \sum_{n \in \mathcal{A}(t)} \sum_{m=1}^{M} \bar{h}_t^n \bar{w}_t^{mn} L(\bar{\alpha}_t^{mn}, \alpha_t^{mn}) - \sum_{t \in I_{\tilde{n}}} \sum_{m=1}^{M} \bar{w}_t^{m\tilde{n}} L(\bar{\alpha}_t^{m\tilde{n}}, \alpha_t^{m\tilde{n}})$$

$$+ \sum_{t \in I_{\tilde{n}}} \sum_{m=1}^{M} \bar{w}_t^{m\tilde{n}} L(\bar{\alpha}_t^{m\tilde{n}}, \alpha_t^{m\tilde{n}}) - \sum_{t \in I_{\tilde{n}}} L(\bar{\alpha}_t^{m^*\tilde{n}}, \alpha^{m^*\tilde{n}}), \tag{32}$$

where $I_{\tilde{n}}$ denotes the time interval during which expert $\tilde{n}$ is active, starting at time $t = \tilde{n}$. The third and fourth terms in the expression are analogous to the regret experienced by expert $\tilde{n}$, as established in Theorem 1. To evaluate the regret for the first and second terms, we employ Lemma 4. The main difference is in the number of experts considered: For a looser bound, assuming that experts remain active beyond their designated lifetime, the maximum number of experts at each time step $t$ would be $gt$. Thus, we derive the following bound for the first and second terms

$$\sum_{t \in I_{\tilde{n}}} \sum_{n \in \mathcal{A}(t)} \sum_{m=1}^{M} \bar{h}_t^n \bar{w}_t^{m\tilde{n}} L(\bar{\alpha}_t^{mn}, \alpha_t^{mn}) - \sum_{t \in I_{\tilde{n}}} \sum_{m=1}^{M} \bar{w}_t^{m\tilde{n}} L(\bar{\alpha}_t^{m\tilde{n}}, \alpha_t^{m\tilde{n}})$$

$$\leq \frac{\sqrt{|I_{\tilde{n}}|} \ln(gt)}{\sigma} + \frac{|I_{\tilde{n}}| \sigma (1+\eta)^2}{\sqrt{|I_{\tilde{n}}|}} \leq \sqrt{|I_{\tilde{n}}|} \left( \ln(gt) + \sigma^2 (1+\eta)^2 \right). \tag{33}$$

By summing (33) with regret bound of Theorem 1 we have

$$\sum_{t \in I_{\tilde{n}}} \sum_{n \in \mathcal{A}(t)} \sum_{m=1}^{M} \bar{h}_t^n \bar{w}_t^{mn} L(\bar{\alpha}_t^{mn}, \alpha_t^{mn}) - \sum_{t \in I_{\tilde{n}}} L(\bar{\alpha}_t^{m^*\tilde{n}}, \alpha^{m^*\tilde{n}})$$

$$\leq \sqrt{|I_{\tilde{n}}|} \left( \frac{(1+2\eta)^2}{2\eta} + \frac{\eta}{2\alpha} + \ln M + (1+\eta)^2 \right) + \sqrt{|I_{\tilde{n}}|} \left( \ln(gt) + \sigma^2 (1+\eta)^2 \right)$$

$$= \sqrt{|I_{\tilde{n}}|} \left( \frac{(1+2\eta)^2}{2\eta} + \frac{\eta}{2\alpha} + \ln M + \ln g + (1+\sigma^2)(1+\eta)^2 + \ln t \right), \tag{34}$$

Until this point, the regret bound we've established applies solely to intervals that start with expert $\tilde{n}$, where each interval's length corresponds to the lifetime of expert $\tilde{n}$. However, to derive a regret bound for any arbitrary time interval $I$, we need to partition the interval into subintervals in a suitable manner. As proposed in [Daniely et al., 2015], we can divide an interval $I$ into two sequences of non-overlapping and consecutive intervals, denoted as $(I_{-p}, ..., I_0)$ and $(I_1, ..., I_q)$, such that $\frac{|I_{r+1}|}{|I_r|} \leq \frac{1}{2}$ for all $r \in (1, q-1)$, and $\frac{|I_r|}{|I_{r+1}|} \leq \frac{1}{2}$ for all $r \in (-p, -1)$. Subsequently, by employing

the inequality $\sum_{r=1}^{\infty} \sqrt{2^{-r}T_0} \leq 4\sqrt{T_0}$, and by replacing $\tilde{n}$ with $n^*$ using (13), we have

$$\sum_{t\in I}\sum_{n\in\mathcal{A}(t)}\sum_{m=1}^{M}\bar{h}_t^n\bar{w}_t^{mn}L(\bar{\alpha}_t^{mn},\alpha_t^{mn}) - \sum_{t\in I}L(\bar{\alpha}_t^{m^*n^*},\alpha^{m^*n^*})$$

$$=\sum_{r=1}^{q-1}\sum_{t\in I_r}\sum_{n\in\mathcal{A}(t)}\sum_{m=1}^{M}\bar{h}_t^n\bar{w}_t^{mn}L(\bar{\alpha}_t^{mn},\alpha_t^{mn}) - \sum_{r=1}^{q-1}\sum_{t\in I_r}L(\bar{\alpha}_t^{m^*n^*},\alpha^{m^*n^*})$$

$$+\sum_{r=-p}^{-1}\sum_{t\in I_r}\sum_{n\in\mathcal{A}(t)}\sum_{m=1}^{M}\bar{h}_t^n\bar{w}_t^{mn}L(\bar{\alpha}_t^{mn},\alpha_t^{mn}) - \sum_{r=-p}^{-1}\sum_{t\in I_r}L(\bar{\alpha}_t^{m^*n^*},\alpha^{m^*n^*})$$

$$\leq A\sqrt{|I|} + B\ln T\sqrt{|I|}. \tag{35}$$

Given that $A$ and $B$ are positive constant variables, we have determined the regret bound for our problem, demonstrating sublinear regret for Algorithm 2.

## A.5  Proof of Lemma 1, Dynamic regret for SAMOCP:

To prove the regret in a dynamic environment we adopt a method which was first proposed by [Besbes et al., 2015]. So we can write the dynamic regret in our problem as

$$\sum_{t=1}^{T}\sum_{n\in\mathcal{A}(t)}\sum_{m=1}^{M}\bar{h}_t^n\bar{w}_t^{mn}L(\bar{\alpha}_t^{mn},\alpha_t^{mn}) - \sum_{t=1}^{T}L(\bar{\alpha}_t^{m^*n^*},\alpha^{m^*n^*})$$

$$+\sum_{t=1}^{T}L(\bar{\alpha}_t^{m^*n^*},\alpha^{m^*n^*}) - \sum_{t=1}^{T}L(\bar{\alpha}_t^{m^*n^*},\alpha_t^{m^*n^*}), \tag{36}$$

where $\alpha_t^{m^*n^*}$ is obtained by (16). The first two terms in (36) represent the static regret as defined in Theorem 3 and have been shown to be bounded. Consequently, to establish the overall regret bound, it is sufficient to find the upper bounds for the third and fourth terms in (36). We begin by dividing the total time interval $T$ into sub-intervals $I_r$ indexed by $r = 1, ..., [T/|I|]$ where $|I|$ is the length of each interval. so we can rewrite (36) as

$$\sum_{r=1}^{[T/|I|]}\sum_{t\in I_r}\sum_{n\in\mathcal{A}(t)}\sum_{m=1}^{M}\bar{h}_t^n\bar{w}_t^{mn}L(\bar{\alpha}_t^{mn},\alpha_t^{mn}) - \sum_{r=1}^{[T/|I|]}\sum_{t\in I_r}L(\bar{\alpha}_t^{m^*n^*},\alpha^{m^*n^*})$$

$$+\sum_{r=1}^{[T/|I|]}\sum_{t\in I_r}L(\bar{\alpha}_t^{m^*n^*},\alpha^{m^*n^*}) - \sum_{r=1}^{[T/|I|]}\sum_{t\in I_r}L(\bar{\alpha}_t^{m^*n^*},\alpha_t^{m^*n^*}). \tag{37}$$

For the second two terms we have

$$\sum_{t\in I_r}L(\bar{\alpha}_t^{m^*n^*},\alpha^{m^*n^*}) - \sum_{t\in I_r}L(\bar{\alpha}_t^{m^*n^*},\alpha_t^{m^*n^*}) \leq 2|I|V(L(.)_{t\in I_r}),$$

where $V(L(.)_t)$ is the variability of environment [Besbes et al., 2015] defined in (14). So the regret in (36) for any arbitrary $|I|$ will be

$$\sum_{t=1}^{T}\sum_{n\in\mathcal{A}(t)}\sum_{m=1}^{M}\bar{h}_t^n\bar{w}_t^{mn}L(\bar{\alpha}_t^{mn},\alpha_t^{mn}) - \sum_{t=1}^{T}L(\bar{\alpha}_t^{m^*n^*},\alpha_t^{m^*n^*})$$

$$\leq \sum_{r=1}^{[T/|I|]}(A+B\ln T)\sqrt{|I|} + (2|I|V(L(.)_{t\in I_r}) = (A+B\ln T)\frac{T}{\sqrt{|I|}} + (2|I|V(L(.)_{t=1}^{T})). \tag{38}$$

Since (35) holds for any interval $I \subseteq [T]$, By selecting $|I| = \left|\left(\frac{T}{V(L(.)_{t=1}^{T})}\right)^{\frac{2}{3}}\right|$ we have

$$\sum_{t=1}^{T}\sum_{n\in\mathcal{A}(t)}\sum_{m=1}^{M}\bar{h}_t^n\bar{w}_t^{mn}L(\bar{\alpha}_t^{mn},\alpha_t^{mn}) - \sum_{t=1}^{T}L(\bar{\alpha}_t^{m^*n^*},\alpha_t^{m^*n^*})$$

$$(A+B\ln T)T^{\frac{2}{3}}V^{\frac{1}{3}}(L(.)_{t=1}^{T}) + 2T^{\frac{2}{3}}V^{\frac{1}{3}}(L(.)_{t=1}^{T}) \leq \tilde{\mathcal{O}}(T^{\frac{2}{3}}V^{\frac{1}{3}}(L(.)_{t=1}^{T})) \tag{39}$$

# B Additional Experiments

## B.1 Synthetic Dataset

In this subsection, we present our analysis using synthetic data to compare our proposed method, SAMOCP, with recent adaptive conformal prediction methods designed for dynamic settings. We analyze our experiments with a new set of learning models to demonstrate how our proposed method can effectively utilize a mixture of learning models to cope with distribution shifts in a dynamic environment with unknown distribution changes. Additionally, experiments with synthetic data also confirm that SAMOCP maintains its advantages when varying the number of learning models. Specifically, we conducted experiments for cases with 2 and 3 learning models.

**Data Generation:** To generate synthetic data that mimics real-world scenarios, we employ two distinct transformation sequences. The first transformation sequence introduces visual noise and blur effects through the application of Gaussian blur and random Gaussian noise. This approach aims to subtly degrade image clarity, simulating real-life challenges such as camera focus issues or atmospheric conditions like fog or mist. The Gaussian blur is applied with moderate settings, while random noise is incorporated to simulate sensor noise or digital compression artifacts commonly encountered in digital imagery. The second transformation sequence focuses on color manipulation. By adjusting image attributes such as brightness, contrast, saturation, and hue in minor increments, we challenge the models to perform reliably under varying lighting conditions and color settings—typical variations that occur due to different times of the day or inconsistencies in camera settings. Additionally, a random conversion of some images to grayscale is employed to further challenge the models' dependency on color information.

In this experiment, two distinct datasets each containing 3000 images are generated from each transformation type. These datasets are designed with a fixed number of 20 classes and hyperparameters $\xi$ and $k_{reg}$ are set to 0.1 and 4, respectively. The variations between the datasets are due to random elements introduced during image processing, such as differences in which pixels are affected by noise or how color properties are altered. This randomness ensures each dataset contains unique instances, even though they stem from the same transformation principles. By concatenating images from the two datasets, the experiment simulates both gradual and sudden distribution shifts. Gradual shifts are seen within the datasets from a single transformation, while sudden shifts occur when switching between datasets from different transformations.

Table 3: Results on the generated synthetic dataset for Efficientnet_b0 and GoogLeNet learning models. The target coverage percentage is 90%. Bold numbers denote the best results in each column for methods with coverage in the $85 - 90$ range. Red numbers indicate unacceptable coverage.

| Model | Method | Coverage (%) | Avg Width | Avg Regret($\times 10^{-3}$) | Run Time |
|---|---|---|---|---|---|
| | SAMOCP | $87.90 \pm 0.26$ | $\mathbf{17.54 \pm 0.01}$ | $0.48 \pm 0.36$ | $19.51 \pm 0.03$ |
| | SAOCP(MM) | $81.98 \pm 7.35$ | $16.54 \pm 1.45$ | $6.97 \pm 2.59$ | $26.30 \pm 0.24$ |
| | | | | | |
| Efficientnet_b0 | FACI | $89.80 \pm 0.31$ | $17.96 \pm 0.05$ | $\mathbf{0.30 \pm 0.18}$ | $6.82 \pm 0.02$ |
| | ScaleFreeOGD | $\mathbf{89.96 \pm 0.00}$ | $18.05 \pm 0.01$ | $1.61 \pm 0.01$ | $\mathbf{2.45 \pm 0.00}$ |
| | SAOCP | $88.73 \pm 0.08$ | $17.82 \pm 0.01$ | $2.18 \pm 0.02$ | $34.43 \pm 0.06$ |
| | | | | | |
| GoogLeNet | FACI | $89.66 \pm 0.30$ | $18.05 \pm 0.09$ | $1.36 \pm 0.79$ | $6.89 \pm 0.02$ |
| | ScaleFreeOGD | $89.95 \pm 0.00$ | $18.07 \pm 0.03$ | $1.82 \pm 0.03$ | $2.46 \pm 0.00$ |
| | SAOCP | $88.39 \pm 0.07$ | $17.75 \pm 0.02$ | $2.55 \pm 0.03$ | $34.78 \pm 0.06$ |

We conducted experiments using a distinct set of learning models. As shown in Table 3, we incorporated two learning models, Efficientnet_b0 and GoogLeNet, where we achieved the smallest prediction set size with coverage close to the target. The SAOCP for GoogLeNet obtained an average width close to our method; however, it is noteworthy that the maximum number of updates in SAOCP is twice that of SAMOCP, demonstrating that our algorithm achieved this result with lower computational costs. Additionally, its regret is almost five times larger than our method's. We also provide synthetic data analysis using another set of learning models consisting of GoogLeNet, DenseNet-121, and EfficientNet-B0 [Tan and Le, 2019], as detailed in Table 4 where our method again was able to construct smaller prediction sets. It should also be noted that, due to severely corrupted data in our synthetic dataset, none of the models were able to produce single width prediction sets that cover true labels.

Table 4: Results on the generated synthetic dataset for EfficientNet-B0, DenseNet-121, and GoogLeNet learning models. The target coverage percentage is $90\%$. Bold numbers denote the best results in each column for methods with coverage in the $85 - 90$ range. Red numbers indicate unacceptable coverage.

| Model | Method | Coverage (%) | Avg Width | Avg Regret($\times 10^{-3}$) | Run Time |
|-------|--------|--------------|-----------|------------------------------|----------|
| | SAMOCP | $88.04 \pm 0.31$ | $\mathbf{17.60 \pm 0.04}$ | $0.65 \pm 0.45$ | $26.39 \pm 0.37$ |
| | SAOCP(MM) | $\color{red}{80.95 \pm 7.75}$ | $16.29 \pm 1.55$ | $7.06 \pm 3.03$ | $34.80 \pm 0.36$ |
| Efficientnet_b0 | FACI | $89.80 \pm 0.31$ | $17.96 \pm 0.05$ | $\mathbf{0.30 \pm 0.18}$ | $7.37 \pm 0.07$ |
| | ScaleFreeOGD | $\mathbf{89.96 \pm 0.00}$ | $18.05 \pm 0.01$ | $1.61 \pm 0.01$ | $\mathbf{2.65 \pm 0.01}$ |
| | SAOCP | $88.73 \pm 0.08$ | $17.82 \pm 0.01$ | $2.18 \pm 0.02$ | $37.25 \pm 0.15$ |
| DenseNet-121 | FACI | $89.72 \pm 0.32$ | $18.04 \pm 0.07$ | $1.07 \pm 0.73$ | $7.42 \pm 0.05$ |
| | ScaleFreeOGD | $89.95 \pm 0.01$ | $18.06 \pm 0.02$ | $1.81 \pm 0.04$ | $2.65 \pm 0.03$ |
| | SAOCP | $88.39 \pm 0.09$ | $17.72 \pm 0.01$ | $2.53 \pm 0.02$ | $37.83 \pm 0.16$ |
| GoogLeNet | FACI | $89.66 \pm 0.30$ | $18.05 \pm 0.09$ | $1.36 \pm 0.79$ | $7.49 \pm 0.06$ |
| | ScaleFreeOGD | $89.95 \pm 0.00$ | $18.07 \pm 0.03$ | $1.82 \pm 0.03$ | $2.66 \pm 0.03$ |
| | SAOCP | $88.39 \pm 0.07$ | $17.75 \pm 0.02$ | $2.55 \pm 0.03$ | $37.90 \pm 0.21$ |

## B.2  TinyImageNet Dataset

Here, we conduct the experiment on a new dataset involving a gradual distribution shift using a new real dataset, TinyImageNet-C, which is a corrupted version of the TinyImageNet [Le and Yang, 2015] dataset that features 200 distinct classes. We have also incorporated a new mixture of learning models—GoogLeNet, DenseNet-121, EfficientNet-B0, and MobileNet-V2 [Li et al., 2021] to demonstrate the performance of our algorithm. In Table 5, we detail our proposed method's comparison with previous methods, where, once again, our algorithm is able to achieve smaller prediction set sizes while maintaining coverage close to the target value of $1 - \alpha$. It is reasonable to study prediction sets in situations where we achieve coverage close to the desired level. The SAOCP(MM) method achieves coverage of less than $85\%$, which is significantly below the target value. Therefore, we do not consider its prediction sets in our comparison. As demonstrated in the table, SAMOCP obtains smaller prediction sets and operates faster than SAOCP. For the TinyImageNet-C dataset, the parameters $\eta$, $\xi$ and $k_{reg}$ are set to 0.025, 0.01 and 20, respectively. Other parameters remain consistent with previous experiments. All real datasets are downloaded from the Zenodo repository.

Table 5: Results on the TinyImageNet-C dataset with a gradual distribution shift. The target coverage is $90\%$, and the average regret is calculated over an interval size of 100. Bold numbers denote the best results in each column for methods with coverage in the 85-90 range. Red numbers indicate unacceptable coverage. SAMOCP achieves the best performance in terms of average width and single width.

| Model | Method | Coverage (%) | Avg Width | Avg Regret($\times 10^{-3}$) | Run Time | Single Width |
|-------|--------|--------------|-----------|------------------------------|----------|--------------|
| | SAMOCP | $87.73 \pm 0.35$ | $\mathbf{171.64 \pm 1.34}$ | $1.28 \pm 0.20$ | $35.22 \pm 0.72$ | 0 |
| | SAOCP(MM) | $\color{red}{84.91 \pm 1.22}$ | $165.93 \pm 3.32$ | $4.80 \pm 0.77$ | $48.55 \pm 0.15$ | 0 |
| GoogLeNet | FACI | $89.69 \pm 0.29$ | $177.93 \pm 1.21$ | $1.07 \pm 0.67$ | $8.48 \pm 0.06$ | 0 |
| | ScaleFreeOGD | $\mathbf{89.96 \pm 0.01}$ | $178.38 \pm 0.49$ | $1.41 \pm 0.08$ | $\mathbf{3.16 \pm 0.03}$ | 0 |
| | SAOCP | $88.36 \pm 0.07$ | $174.81 \pm 0.50$ | $2.02 \pm 0.08$ | $40.83 \pm 0.32$ | 0 |
| DenseNet-121 | FACI | $89.68 \pm 0.31$ | $176.66 \pm 1.10$ | $1.30 \pm 0.74$ | $8.54 \pm 0.11$ | 0 |
| | ScaleFreeOGD | $89.95 \pm 0.02$ | $177.24 \pm 0.64$ | $1.53 \pm 0.06$ | $3.18 \pm 0.05$ | 0 |
| | SAOCP | $88.48 \pm 0.12$ | $173.79 \pm 0.57$ | $2.09 \pm 0.04$ | $40.09 \pm 0.36$ | 0 |
| Efficientnet_b0 | FACI | $89.64 \pm 0.35$ | $176.78 \pm 1.03$ | $1.01 \pm 0.61$ | $8.56 \pm 0.07$ | 0 |
| | ScaleFreeOGD | $89.95 \pm 0.01$ | $177.23 \pm 0.63$ | $1.37 \pm 0.06$ | $3.17 \pm 0.05$ | 0 |
| | SAOCP | $88.30 \pm 0.18$ | $173.52 \pm 0.63$ | $1.98 \pm 0.08$ | $41.06 \pm 0.15$ | 0 |
| Mobilenet_v2 | FACI | $89.69 \pm 0.30$ | $175.26 \pm 0.95$ | $\mathbf{0.98 \pm 0.52}$ | $8.55 \pm 0.07$ | 0 |
| | ScaleFreeOGD | $89.95 \pm 0.01$ | $175.82 \pm 0.51$ | $1.36 \pm 0.07$ | $3.19 \pm 0.04$ | 0 |
| | SAOCP | $88.30 \pm 0.15$ | $171.83 \pm 0.55$ | $1.94 \pm 0.06$ | $40.60 \pm 0.23$ | 0 |

### B.3 SAMOCP vs. SAOCP (equal lifetimes)

We compared our method with a specific version of SAOCP where the hyperparameter $g$ in equation (9) is set to 8, giving both SAOCP and SAMOCP experts the same lifetime. We conduct experiments identical to those described in section 4, addressing both sudden and gradual shifts for CIFAR-10C and CIFAR-100C, respectively. Our results are demonstrated in Table 6 for CIFAR-10C and in Table 7 for CIFAR-100C. For both cases, we observe the advantage of utilizing multiple learning models instead of a single model, as our proposed method, SAMOCP, significantly outperforms SAOCP across all four learning models. SAMOCP achieved better results in terms of coverage, average width, average regret, and single width compared to SAOCP.

Table 6: Performance evaluation of SAOCP with same lifetime time as SAMOCP for sudden distribution shift setting on CIFAR-10C.

| Model | Method | Coverage (%) | Avg Width | Avg Regret($\times 10^{-3}$) | Run Time | Single Width |
|---|---|---|---|---|---|---|
| | SAMOCP | **88.37 $\pm$ 0.23** | **1.24 $\pm$ 0.06** | **0.98 $\pm$ 0.11** | 33.75 $\pm$ 0.34 | **0.69 $\pm$ 0.03** |
| DenseNet-121 | SAOCP | 87.16 $\pm$ 0.10 | 1.45 $\pm$ 0.03 | 4.36 $\pm$ 0.14 | **14.73 $\pm$ 0.09** | 0.53 $\pm$ 0.01 |
| ResNet-50 | SAOCP | 87.87 $\pm$ 0.15 | 1.52 $\pm$ 0.01 | 3.91 $\pm$ 0.11 | 14.78 $\pm$ 0.07 | 0.52 $\pm$ 0.01 |
| ResNet-18 | SAOCP | 87.16 $\pm$ 0.18 | 1.51 $\pm$ 0.02 | 4.38 $\pm$ 0.16 | 14.85 $\pm$ 0.08 | 0.51 $\pm$ 0.01 |
| GoogLeNet | SAOCP | 87.74 $\pm$ 0.10 | 1.45 $\pm$ 0.01 | 3.92 $\pm$ 0.15 | 14.84 $\pm$ 0.06 | 0.54 $\pm$ 0.01 |

Table 7: Performance evaluation of SAOCP with same lifetime time as SAMOCP for gradual distribution shift setting on CIFAR-100C.

| Model | Method | Coverage (%) | Avg Width | Avg Regret($\times 10^{-3}$) | Run Time | Single Width |
|---|---|---|---|---|---|---|
| | SAMOCP | **88.16 $\pm$ 0.18** | **5.43 $\pm$ 0.28** | **0.92 $\pm$ 0.07** | 34.87 $\pm$ 0.67 | **0.29 $\pm$ 0.01** |
| DenseNet-121 | SAOCP | 87.33 $\pm$ 0.14 | 6.12 $\pm$ 0.16 | 4.03 $\pm$ 0.06 | **14.89 $\pm$ 0.06** | 0.27 $\pm$ 0.00 |
| ResNet-50 | SAOCP | 87.22 $\pm$ 0.12 | 6.75 $\pm$ 0.16 | 3.99 $\pm$ 0.14 | 15.01 $\pm$ 0.07 | 0.26 $\pm$ 0.00 |
| ResNet-18 | SAOCP | 87.15 $\pm$ 0.14 | 7.24 $\pm$ 0.22 | 4.10 $\pm$ 0.11 | 15.02 $\pm$ 0.08 | 0.25 $\pm$ 0.00 |
| GoogLeNet | SAOCP | 87.09 $\pm$ 0.16 | 6.78 $\pm$ 0.12 | 4.14 $\pm$ 0.13 | 14.98 $\pm$ 0.09 | 0.26 $\pm$ 0.00 |

### B.4 SAMOCP vs. MOCP

To validate the advantage of SAMOCP versus MOCP in dynamic environments, we conducted experiments for both sudden and gradual distribution shifts. see Tables 8 and 9. In both tables, SAMOCP outperforms MOCP in terms of Average Width and Single Width metrics.

Table 8: Comparison of MOCP and SAMOCP on the CIFAR-10C dataset with a sudden distribution shift. The target coverage is 90%. Bold numbers denote the best results in each column.

| Method | Coverage (%) | Avg Width | Single Width |
|---|---|---|---|
| MOCP | **89.96 $\pm$ 0.33** | 1.29 $\pm$ 0.08 | 0.67 $\pm$ 0.04 |
| SAMOCP | 88.37 $\pm$ 0.23 | **1.24 $\pm$ 0.06** | **0.69 $\pm$ 0.03** |

Table 9: Comparison of MOCP and SAMOCP on the CIFAR-100C dataset with a gradual distribution shift. The target coverage is 90%. Bold numbers denote the best results in each column.

| Method | Coverage (%) | Avg Width | Single Width |
|--------|-------------|-----------|--------------|
| MOCP | **89.77 ± 0.19** | 5.85 ± 0.28 | 0.28 ± 0.01 |
| SAMOCP | 88.16 ± 0.18 | **5.43 ± 0.28** | **0.29 ± 0.01** |

