# OpenReview forum: "Multi-model Ensemble Conformal Prediction in Dynamic Environments"
_NeurIPS.cc/2024/Conference — NeurIPS 2024 poster_

### Official Review · Reviewer_4Km4 · 2024-06-27

**Soundness:** 3
**Presentation:** 3
**Contribution:** 2
**Rating:** 5
**Confidence:** 4

**Summary:**

This paper considers conformal prediction in the online setting with multiple fitted models, which is very interesting and in line with real applications. As it remains an active question regarding how to aggregate multiple models, the proposed approach adopts importance sampling to choose the 'best' model at each step. The performance is justified with comprehensive simulations and comparisons with baseline approaches are presented.

**Strengths:**

The problem studied in this paper is well-defined and is of interest in the literature. The theoretical guarantee and numerical performance of the proposed approach are well presented.

**Weaknesses:**

Please see questions in the following section.

**Questions:**

1. [Choice of weights] The weights defined in (6) follow exponential decay. It would be interesting to explore other choices of the weights: (1) in terms of theoretical guarantee, how will the coverage/accuracy guide the choice of w? (2) choices of w can also be compared via numerical simulations.

2. [Thm 1] Average regret is considered in Thm 1. It remains unknown how will the MOCP approach outperform FACI in terms of the coverage guarantee.

3. [MOCP vs SAMOCP] MOCP would also be capable of the tasks presented in the simulation section, then it could be more informative to compare SAMOCP with MOCP to validate the advantage of SAMOCP.

4. [From adversarial setting to stochastic setting] It would be interesting to study the theoretical performance of the proposed approach under the stochastic setting where observations are i.i.d. to understand the gains of model aggregation. Please see references e.g. (1) Duchi et al. (Adaptive Subgradient Methods for Online Learning and Stochastic Optimization); (2) Hazan et al. (Adaptive Online Gradient Descent); etc.

**Limitations:**

Limitations are discussed in the paper.

---

> ### Author Rebuttal · Authors · 2024-08-07
>
> Q1: The exponential update framework is theoretically justified by its ability to minimize regret, a measure of how well the algorithm performs relative to the best-fixed strategy in hindsight. The exponential update is widely used in many online setting algorithms, see [1,2,3].
> Furthermore, the choice of such an exponential update is rooted in the principles of online mirror descent in the case where the variables are constrained to a probability simplex (non-negative and sum up to one). Hence, it naturally fits the ideal weight update.
> That said, we would be happy to evaluate alternative weight update methods that the reviewer may suggest.
>
>
> Q2: We understand the concern. To address this comment, we conduct additional experiments comparing MOCP and FACI methods in terms of Coverage and Regret, see Tables 1 and 2 in the attached file. These experiments involve sudden distribution shifts with CIFAR-10C and gradual distribution shifts with CIFAR-100C, respectively. In both cases, MOCP outperformed FACI, demonstrating the advantage of using multiple learning models over a single model.
> While we did not theoretically analyze the coverage for MOCP, we provided the corresponding analysis guarantee for SAMOCP as shown in Theorem 2. The coverage of FACI was studied for a special case where the step size is diminishing. As mentioned in [1], “We evaluate the long-term coverage in a specialized case where the hyperparameters decay to 0 over time.” However, to provide a coverage guarantee, we did not need to consider such a special case.
>
>
> Q3: Thank you for your suggestion. We conducted experiments to compare MOCP and SAMOCP for both sudden and gradual distribution shifts, see Tables 3 and 4 in the attached file. In both tables, SAMOCP outperforms MOCP in terms of Average Width and Single Width metrics.
>
>
> Q4: We appreciate the reviewer's suggestion to study the theoretical behavior of our approach under the stochastic setting where observations are i.i.d. We agree with the reviewer that it would indeed be an interesting future direction of the present work.  We will cite the references and mention this future direction in the final version.
>
> [1] I. Gibbs and E. Candès, “Conformal inference for online prediction with arbitrary distribution shifts.”
>
> [2] Y. Shen et al. “Random Feature-Based Online Multi-Kernel Learning in Environments with Unknown Dynamics.”
>
> [3] Elad Hazan, "Introduction to Online Convex Optimization."

---

> > ### Comment · Reviewer_4Km4 · 2024-08-10
> > **Follow-up comments**
> >
> > Thank you for the references and additional simulation studies! I'll maintain my scores at this moment and keep the authors' responses in mind for my evaluation through the discussion period.

---

### Official Review · Reviewer_76a7 · 2024-07-12

**Soundness:** 2
**Presentation:** 3
**Contribution:** 2
**Rating:** 5
**Confidence:** 4

**Summary:**

This paper proposes an online multimodal conformal prediction method named SAMOCP, developed to address data distribution shifts in dynamic environments. Specifically, the method selects the best model from multiple candidates to create prediction sets via strongly adaptive online learning.

**Strengths:**

1. The method aims to expand single-modal online conformal prediction to multimodal.
2. The method achieves better efficiency through the selection from multiple models.

**Weaknesses:**

1. This paper appears to be a variant of SAOCP [1], utilizing many techniques from [1] to construct candidate sets with multiple models. By optimizing the weight of each model, the final prediction set is obtained from the best model. However, there is insufficient evidence to validate the superiority of SAMOCP compared to SAOCP, both theoretically and empirically.
2. The authors claim that the method is devised for a multimodal setting. However, both the methodology and experiments seem to involve multiple experts for a unimodal dataset using different deep neural networks, which contradicts the introduction of the method.
3. There are several concerns regarding the theoretical analyses. What is the meaning of Theorems 1 and 3? Are they intended to demonstrate that the pinball loss will increase with the addition of the optimal single model? What is the strength of these theoretical analyses for decision-making in an online multimodal setting?
4. It is suggested to provide a Complexity Analysis of the overall model.

**Questions:**

See in the Weaknesses.

**Limitations:**

The limitations of the authors have been pointed out in the Conclusion.

---

> ### Author Rebuttal · Authors · 2024-08-06
>
> We sincerely thank the reviewer for reading our manuscript and thoughtful comments and questions. We will address your questions as follows:
>
> Q1: We would like to clarify that our proposed method, SAMOCP, is not simply a multimodal version of SAOCP [1]. SAMOCP includes a distinct approach for updating the weight assigned to each expert and having a specific step size for each expert, which is different from SAOCP. As a result, SAMOCP demonstrates superior performance compared to both the original SAOCP and the multimodal version of SAOCP (referred to as SAOCP(MM)). Specifically, SAMOCP outperforms SAOCP across multiple metrics, including Average Width, Average Regret, Run Time, and Single Width, see Tables 1 and 2. Meanwhile,  SAMOCP achieves similar coverage as SAOCP, with < 1% difference. This indicates that SAMOCP maintains the desired coverage while achieving superior performance in other metrics.
> In addition, we conducted comprehensive experiments to compare SAMOCP with SAOCP. In these experiments, the lifetimes of experts were same in both methods, with an equal parameter g in Equation (9). The detailed results of these experiments are presented in Tables 6 and 7. These comparisons validate that SAMOCP consistently achieves better performance in terms of Coverage, Average Width, Average Regret, and Single Width.
>
> Q2: There might be some misunderstanding. The `multimodel’ does not mean the dataset contains multiple modalities. Instead, it refers to the fact that the proposed SAMOCP employs multiple learning models (neural networks in our experiments).
> The intuition behind the multimodal design is that different learning models exhibit varying behaviors in response to distinct distribution shifts. A single learning model may not consistently provide the best performance across different distribution shifts. By leveraging multiple learning models, SAMOCP dynamically selects the suitable one for the current data distribution.
>
> Q3: Theorems 1 and 3 aim to demonstrate that the proposed method achieves sublinear regret, which bounds the difference between the loss of the proposed online algorithm and that of the best model in hindsight. While the cumulative difference increases over time, sublinear regret indicates that the average difference decreases over time, and converges to zero as time goes to infinity, and henceforth achieves “no regret” on average. Sublinear regret has been a crucial criterion in analyzing various online learning frameworks, see [2,3,4,5,6]. Below we will elaborate in further detail about the meaning of Theorem 1 and 3.
>
> Theorem 1 shows that the Multimodal Online Conformal Prediction (MOCP) algorithm achieves sublinear regret in static environments. Specifically, Theorem 1 analyzes the difference in loss between the MOCP algorithm and the best model in hindsight. This means that as the number of time steps T increases, the average difference in performance between the MOCP algorithm and the best possible model decreases, indicating that MOCP adapts well to the data over time.
>
> Theorem 3 provides the regret analysis for the Strongly Adaptive Multimodal Online Conformal Prediction (SAMOCP) algorithm. Specifically, Theorem 3 evaluates the difference in loss between the SAMOCP algorithm and the best expert (where each expert selects the optimal model) in hindsight over any intervals with arbitrary length. This analysis shows that the loss difference between the SAMOCP method and the best expert converges to zero as time goes to infinity. This indicates that SAMOCP can effectively select the most suitable expert from multiple candidates.
>
> Q4: For the first proposed algorithm (MOCP), considering there are M models, the overall computational complexity is O(MT). For the SAMOCP algorithm, given that the maximum number of active experts at each time step g[log T], the overall computational complexity is O(MgTlog T).
>
>
> [1] A. Bhatnagar et al. "Improved Online Conformal Prediction via Strongly Adaptive Online Learning."
>
> [2] S. Paternain et al. "Constrained Online Learning in Networks with Sublinear Regret and Fit."
>
> [3] Y. Zhou et al. "Regret Bounds Without Lipschitz Continuity: Online Learning With Relative-Lipschitz Losses."
>
> [4] C.A. Cheng et al. "Online Learning With Continuous Variations: Dynamic Regret and Reductions."
>
> [5] S. Ito et al. "Efficient Sublinear-Regret Algorithms for Online Sparse Linear Regression With Limited Observation"
>
> [6] S. Paternain et al. "Constrained Online Learning in Networks with Sublinear Regret and Fit."

---

> ### Comment · Reviewer_76a7 · 2024-08-13
> **Question**
>
> I want to thank the authors for their responses and clarifications. I have read through all of them and am willing to increase my score. However, my confusion still persists: why is the method named "Multimodal Conformal Prediction"? In my view, there is a significant difference between "multimodel" and "multimodal." "Multimodel" typically refers to the use of multiple models, often based on ensemble learning, whereas "multimodal" pertains to multiple types of data, which should account for the interaction between different modalities, leading to different methods or theoretical assumptions.

---

> > ### Author Response · Authors · 2024-08-14
> > **Official Comment by Authors**
> >
> > Thank you for your willingness to increase the scores based on our response, and for pointing out the confusion regarding "multimodal" and "multimodel." We understand the concern. According to your comment, we will change "multimodal" to "multimodel" to avoid confusion. We appreciate your insightful comment, and we agree that this change will help improve the clarity of the paper. We would be happy to further clarify any remaining questions or concerns you may have. Please let us know if you have further questions or concerns.

---

### Official Review · Reviewer_xzo3 · 2024-07-12

**Soundness:** 3
**Presentation:** 3
**Contribution:** 3
**Rating:** 5
**Confidence:** 4

**Summary:**

This paper investigates online conformal prediction within dynamic environments. The authors introduce a novel adaptive conformal prediction framework that leverages multiple candidate models. The proposed algorithm achieves sublinear regret, and its effectiveness is demonstrated through both real and synthetic datasets.

**Strengths:**

The proposed method is novel, with a clear presentation and comprehensive literature review. It demonstrates good performance compared to benchmark methods.

**Weaknesses:**

The intuition behind the proposed algorithm in the dynamic environment (Section 3.2) is not sufficiently developed.

**Questions:**

1. In Algorithm 1, how should \(\epsilon\) be chosen?

2. How does this multi-model approach perform compared to retraining the non-conformal score after each update?

3. How quickly can this method adapt to distribution shifts compared to benchmarks in real data studies? Rapidly capturing distribution shifts and making valid prediction sets is crucial in practical applications.

4. Could the authors provide intuition behind the design of the algorithm in a dynamic environment? Specifically, why are many experts retained when \(t\) is large, despite having a total of \( M\) models?

---

> ### Author Rebuttal · Authors · 2024-08-07
>
> The authors would like to thank the reviewer for their valuable comments and for recognizing the novelty of our work. We will address your questions as follows:
>
> Q1: The parameter $\epsilon$ was selected via grid search from {0.1, 0.2, …,0.9}.  The one which led to the smallest prediction set size (Avg Width) while ensuring desired coverage was chosen, and $\epsilon= 0.9$ in all our experiments.
>
> Q2: We would like to clarify that the non-conformal scores are indeed calculated after each update. Specifically,  upon receiving each new data sample, the non-conformity scores for each candidate label are calculated via eq (17). The calibration set is then updated, once the true label is received.
>
> Q3: Our experimental results show that SAMOCP can adapt to distribution shift faster than the benchmarks in real data sets, see Figure 2, which shows that SAMOCP consistently achieves lower regret with different window sizes. Note that a lower regret implies that the algorithm adapts faster to changes. In addition, the regret calculated over a smaller window size characterizes the algorithm’s adaptivity in a shorter time interval, hence faster in capturing distribution shifts. Therefore, Figure 2 indicates that the SAMOCP can adapt faster to distribution shifts compared to benchmarks in various time scales. Thanks for your comments, we will revise the final version to provide more intuition of this experimental results.
>
> Q4: Thank you for your question. The intuition behind the SAMOCP algorithm is to effectively adapt to dynamic environments with unknown distribution shifts. SAMOCP leverages multiple variants of the MOCP algorithm, each treated as an expert with specific values for stepsize $\epsilon$  and lifetime. This design allows the algorithm to maintain a diverse set of experts, each suited to a different type of shift. Experts with shorter lifetimes can rapidly adapt to quick distribution shifts. On the other hand, experts with longer lifetimes are more suitable for tracking gradual or subtle changes.
> In addition, we would like to clarify that the number of experts is not influenced by the number of models in SAMOCP. Instead, SAMOCP treats each instance of MOCP as an expert with a specific lifetime. The reason many experts are retained for large t is that more experts become activated as time increases, and experts with different lengths of lifetime are incorporated to cope with different types of distribution shifts. Such a mechanism enables the SAMOCP to be strongly adaptive to unknown distribution shifts and achieve sub-linear regret.

---

### Official Review · Reviewer_moxN · 2024-07-17

**Soundness:** 3
**Presentation:** 3
**Contribution:** 3
**Rating:** 6
**Confidence:** 4

**Summary:**

This paper proposes the Strongly Adaptive Multimodal Online Conformal Prediction (SAMOCP) methodology, which constructs adaptive conformal prediction sets by integrating information from multiple learning models in dynamic environments. This is accomplished by creating multiple experts at each time step, where each expert is an online learning algorithm that dynamically updates the weights of each model and has a finite active time interval. The appropriate miscoverage probability at time $t$ for constructing the prediction set in the subsequent step is determined by the miscoverage probability associated with the selected expert among the many active experts.

**Strengths:**

1. The concept of developing a Conformal Prediction framework for dynamic environments using information from multiple learning models is novel.

2. The paper is well-written and well-structured.

3. The performance of the proposed method is analyzed with theoretical justifications and has demonstrated advantages through extensive experiments compared to existing benchmarks.

**Weaknesses:**

1. As already pointed out in this paper, this new method is much slower compared to other benchmarks. Therefore, it would be beneficial if the authors could conduct a comprehensive computational complexity analysis, in addition to reporting the empirical runtime in the experiments. This would help practitioners understand how well this method can scale to more complex datasets with a larger number of learning models.

**Questions:**

1. Did you try to implement the method on time series data where consecutive data can potentially exhibit continuous distributional shifts? For example, on stock price data, with the response output being 1 if the price goes up and 0 if it goes down.

2. Can you explain how exactly to ‘select one of the miscoverage probabilities ${\alpha_t^m\}$ according to the pmf …’? Is it simply selecting the $\alpha_t^m$ with the highest normalized weight?

3. In the numerical experiments, if my understanding is correct, the sudden shifts setting involves having intact data before a certain threshold and fully corrupted data after that threshold. How exactly does it work for the gradual shifts? For instance, if there are five distinct levels of severity, do you assign all images with level 0 in the first 1/5 of the horizon, level 1 in 1/5-2/5 of the horizon, and so on?

4. How is the number of experts chosen?

Minor:
There seems to be a typo in line 134: the miscoverage probability $\alpha_{t+1}^m$ can be updated via ..., instead of $\alpha_{t}^m$.

For completeness and better clarity, I would suggest adding a line in the algorithm between ‘select one miscoverage probability…’ and ‘Observe true $Y_{t+1}$' that says ‘construct prediction set for $Y_{t+1}$ using the selected miscoverage probability.'

**Limitations:**

The authors addressed the limitations of their work.

---

> ### Author Rebuttal · Authors · 2024-08-07
>
> Thank you for your time and effort in reviewing our submission.
>
> Q1: We did not conduct experiments with time series data. According to the dynamic regret analysis in Lemma 1, achieving sub-linear dynamic regret requires the variation of the loss function to be sublinear. Having continuous distribution shifts in consecutive data violates this requirement. Hence, experiments in this setting were not included.
>
> Q2: The miss coverage probability  $\alpha_t^m$​ is not simply selected based on the highest probability. Instead, each miss coverage probability $\alpha_t^m$ is chosen with probability proportional to its normalized weight $\bar{w}_t^m$. In other words, miss coverage probabilities of models with higher weights are more likely to be selected. Such probabilistic selection enables the algorithm to consider a variety of models.
>
> Q3: For every gradual shift experiment (e.g. Table 1 in section 4), the data sequence was split into batches of 500 data samples each. The severity changes (increases or decreases) after each batch of data. The severity starts at level 0 and increases one by one after each batch until it reaches level 5. After reaching level 5, the severity decreases one by one and goes back to level 0 in subsequent batches. This cycle of increasing and decreasing severity continues throughout the duration of the experiment. To better illustrate the procedure of the gradual shift, please refer to the following sequence, which shows the change of severity through gradual distribution shifts.
> 0->1->2->3->4->5->4->3->2->1->0->1->....
> This setup indicates that the data experiences gradual shifts in both directions, providing a robust evaluation of the algorithm’s adaptability.
>
> Q4: The number of experts is not chosen directly. Instead, the lifetime of each expert is determined by the formula in Equation (9). This formula depends on the parameter g. To determine the value of g, we employed a grid search approach within the candidates {4,8,16,24,32,48,64}. The parameter g that led to the smallest prediction set size (Avg Width) while maintaining reasonable coverage and runtime was selected, which was g=8.
>
> Computational Complexity: For the first proposed algorithm (MOCP), considering there are M models, the overall computational complexity is O(MT). For the SAMOCP algorithm, given that the maximum number of active experts at each time step g[log T], the overall computational complexity is O(MgTlog T).
>
> We sincerely thank the reviewer for the insightful suggestion. We included your suggestion in our algorithm to improve clarity and completeness.

---

> > ### Comment · Reviewer_moxN · 2024-08-10
> >
> > Thank you to the authors for the detailed responses. I am happy to maintain my current score and have increased my confidence in it.

---

### Author Rebuttal · Authors · 2024-08-07

We would like to thank all the reviewers for their time and effort in reviewing our paper.

Additional experiments have been conducted to compare our first algorithm (MOCP) with benchmarks and SAMOCP. The results can be found in the attached file.

---

### Decision · Program_Chairs · 2024-09-25

**Decision:**

Accept (poster)

**Comment:**

The paper proposes an adaptive conformal prediction algorithm that integrates predictions from multiple learning models in a dynamic environment. The paper provides both theoretical and empirical results.
The paper received four reviews, and after the rebuttal all reviewers are leaning towards accepting the paper. The reviewers note that the proposed algorithm is novel and the theory and the experiments are sound and sufficient. During he rebuttal, the reviewers had a few concerns and questions, including regarding the interpretation of the theorems, the speed of the method, and hyperparameter selection. Those concerns were addressed during the rebuttal, and no major concerns remain.
One reviewer pointed out that the usage of `multimodal' (including the title) throughout should be changed to `mutimodel', and I agree.
Overall this is a valuable contribution and good addition to the conference program.